# PLASTIC: Improving Input and Label Plasticity for Sample Efficient Reinforcement Learning

**Hojoon Lee*   Hanseul Cho*   Hyunseung Kim*   Daehoon Gwak   Joonkee Kim**
**Jaegul Choo   Se-Young Yun   Chulhee Yun**
Kim Jaechul Graduate School of AI, KAIST
{joonleesky, jhs4015, mynsng, daehoon.gwak, joonkeekim,
jchoo, yunseyoung, chulhee.yun}@kaist.ac.kr

## Abstract

In Reinforcement Learning (RL), enhancing sample efficiency is crucial, particularly in scenarios when data acquisition is costly and risky. In principle, off-policy RL algorithms can improve sample efficiency by allowing multiple updates per environment interaction. However, these multiple updates often lead the model to overfit to earlier interactions, which is referred to as the *loss of plasticity*. Our study investigates the underlying causes of this phenomenon by dividing plasticity into two aspects. *Input plasticity*, which denotes the model's adaptability to changing input data, and *label plasticity*, which denotes the model's adaptability to evolving input-output relationships. Synthetic experiments on the CIFAR-10 dataset reveal that finding smoother minima of loss landscape enhances input plasticity, whereas refined gradient propagation improves label plasticity. Leveraging these findings, we introduce the **PLASTIC** algorithm, which harmoniously combines techniques to address both concerns. With minimal architectural modifications, PLASTIC achieves competitive performance on benchmarks including Atari-100k and Deepmind Control Suite. This result emphasizes the importance of preserving the model's plasticity to elevate the sample efficiency in RL. The code is available at https://github.com/dojeon-ai/plastic.

## 1 Introduction

In Reinforcement Learning (RL), achieving sample efficiency is crucial in various domains including robotics, autonomous driving, and healthcare, where data acquisition is constrained and expensive [42, 36]. In theory, off-policy RL algorithms promise increased sample efficiency by allowing multiple updates of policy or value functions from a single data instance [26, 24, 43]. However, they tend to suffer from the *loss of plasticity*, a phenomenon where the models overfit to earlier interactions and fail to adapt to new experiences [52, 19, 43].

The origins of the loss of plasticity are a focus of contemporary research. One avenue of study points is the role of the smoothness of the loss surface. Models seeking smoother minima of loss landscape tend to exhibit more stable learning patterns and enhanced plasticity [46, 47].

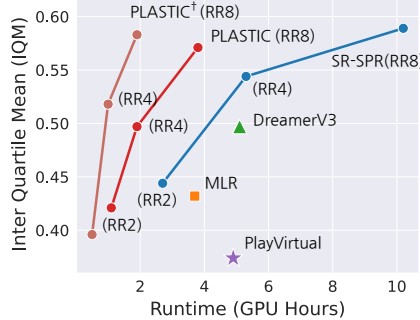

Figure 1: Scaling of our proposed method, PLASTIC, on the Atari-100k benchmark compared to state-of-the-art methods (Section 4.3).

---

*Equal contributions.

37th Conference on Neural Information Processing Systems (NeurIPS 2023).

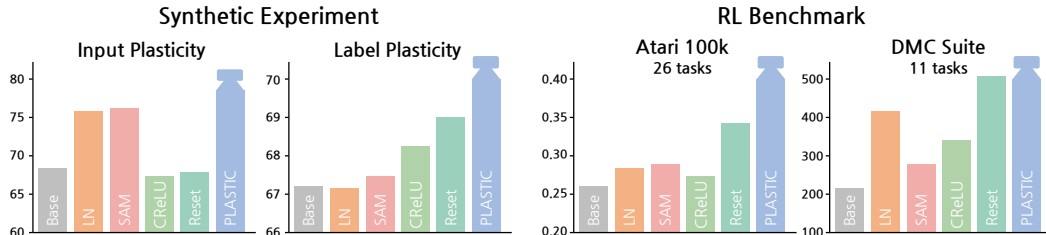

Figure 2: **Left:** Performance of Synthetic Experiments. Layer Normalization (LN) and Sharpeness-Aware Minimization (SAM) considerably enhance input plasticity, while their effect on label plasticity is marginal. Conversely, Concatenated ReLU (CReLU) and periodic reinitialization (Reset) predominantly improve label plasticity with subtle benefits on input plasticity. **Right:** Performance of RL Benchmarks. PLASTIC consistently outperforms individual methods, highlighting the synergistic benefits of its integrated approach.

Another line of investigation emphasizes the role of gradient propagation, highlighting that as training continues, the saturation of active units can impede weight updates, diminishing the model's adaptability [18, 2].

To understand the interplay of these factors within the context of RL, we designed a couple of synthetic experiments. The RL agent often confronts two adaptation scenarios, which involve adjusting to new data inputs and evolving input-label dynamics. In particular, an example of the latter might be an action once considered to be positive but later perceived as negative. Therefore, throughout this work including our synthetic experiments, we categorize the model's plasticity into two cases:

- **Input Plasticity:** adaptability to shifts in input data distributions, $p(x)$.

- **Label Plasticity:** adaptability to evolving labels (returns) for given inputs (observations), $p(y|x)$.

For our exploration, we employ the CIFAR-10 dataset [37]. To assess input plasticity, we train the model from a progressively growing dataset, where the data is segmented into 100 chunks and sequentially added to the buffer. On the other hand, label plasticity is examined by altering labels at intervals, reflecting the dynamic nature of input-output associations in RL. We also adopt strategies to enhance loss smoothness and gradient propagation in the tested models. The Sharpeness-Aware Minimization (SAM) optimizer [21] and Layer Normalization (LN) [5] are employed for the former, while periodic reinitialization of the last few layers (Reset) [81, 52] and Concatenated ReLU (CReLU) activations [61, 2] are used to enhance the latter.

Our synthetic experiments yield clear distinctions between the proposed two concepts of plasticity. As illustrated on the left side of Figure 2, smoother minima of loss largely improve input plasticity, while maintaining the amount of gradient propagation has a pronounced impact on label plasticity. Building on these findings, we propose an algorithm called **PLASTIC**, which combines SAM optimizer, LN, periodic Reset, and CReLU activation to improve both input and label plasticity. The simplicity of PLASTIC facilitates its seamless integration into standard off-policy RL frameworks, requiring minimal code modifications. Notably, across the Atari-100k and Deepmind Control Suite benchmarks, PLASTIC achieved competitive performance to the leading methods, underscoring its potential in enhancing RL's sample efficiency.

In summary, our main contributions are listed as follows:

- Through synthetic experiments, we find out that loss smoothness and refined gradient propagation play separate roles in improving the model plasticity (Section 3).

- We introduce the PLASTIC, a simple-to-use and efficient algorithm that improves the model's plasticity by seeking a smooth region of loss surface and preserving gradient propagation.

- Empirically, PLASTIC achieved competitive performance on challenging RL benchmarks, including Atari-100k and Deepmind Control Suite (Section 4.2-4.5).

## 2 Preliminaries

### 2.1 Sample Efficient Reinforcement Learning

Achieving sample efficiency is crucial for applying RL to real-world problems. Constraints such as limited online data collection (e.g., robotics, educational agents) or safety considerations (e.g., autonomous driving, healthcare) emphasize the need for efficient RL algorithms [36, 42].

Off-policy RL algorithms like Rainbow [26] and SAC [24] offer improved sample efficiency by permitting multiple policy or value function updates based on a single data collection. However, this advantage may introduce a pitfall: the increased update rate can lead to overfitting, undermining the model's generalizability and adaptability to new datasets [29, 43].

To overcome these challenges, various methods have been proposed:

- Data Augmentation: Renowned for its effectiveness in computer vision [16, 80], data augmentation excels in visual RL algorithms [73, 72, 40, 55, 68], particularly when combined with self-supervised learning methods [39, 58, 77].
- Regularization: Diverse regularization strategies have also demonstrated their effectiveness, including L2 regularization [45], spectral normalization [23, 11], dropout [27], and both feature mixing [12] and swapping [10].
- Self-Supervised Learning: By incorporating auxiliary learning objectives, self-supervised learning has emerged as a potential solution. Objectives encompass pixel or latent space reconstruction [74, 77], future state prediction [58, 19, 41], and contrastive learning focusing on either instance [39, 20] or temporal discrimination [53, 64, 48].

Despite these advancements, a key question remains. Why does an overfitted model face challenges when adapting to new datasets?

### 2.2 Understanding Plasticity in Reinforcement Learning

The question for understanding model plasticity in RL is driven by the inherent need for plasticity as agents consistently encounter new inputs and evolving input-output relationships.

A seminal contribution by Lyle et al. [47] highlighted the importance of smoother loss landscapes. By applying layer normalization across all convolutional and fully connected layers, they managed to flatten the loss landscapes, subsequently enhancing performance in Atari environments. While the emphasis on smooth loss landscapes is relatively new in RL, its importance has been substantiated in supervised learning, where empirical evidence suggests models converged on a wider and smoother loss surface generalize better to unseen datasets [34, 30, 71]. Following this insight, the sharpness-aware minimization (SAM) optimizer [21] has recently gained attention in supervised learning [13, 44], aiming to minimize both training loss and its sharpness.

Concurrently, various studies point out a progressive decline in a network's active units as a probable cause for the loss of plasticity [2, 18]. As neural networks iteratively adjust their weights to minimize training losses, the number of active units tends to shrink, often culminating in the dead ReLU phenomenon [28, 17]. This reduction in active units hampers the gradient propagation to upper layers, thus impairing network adaptability. Proposed remedies include periodic neuron reinitialization [52, 19, 63] or employing Concatenated ReLU activation [61, 2], both showing promise in RL.

With these observations in mind, our primary endeavor is to delve into the complex interplay of these factors. We aim to analyze whether these factors synergistically affect the model's plasticity or operate as distinct, individual contributors.

### 2.3 Off-Policy Reinforcement Learning Algorithms

**Rainbow.** Rainbow [26] is a widely used off-policy algorithm for discrete control settings that integrates six different extensions to the standard DQN algorithm [50]. The extensions include Double Q-learning [66], Prioritized Experience Replay [57], Dueling Networks [69], Multi-Step Return, Distributional Q-function [8], and Noisy Networks [22]. Rainbow significantly improves the performance and robustness of the standard DQN algorithm, thereby addressing shortcomings in function approximation and exploration-exploitation trade-offs.

The Q-value updates in the Rainbow algorithm follow the principle of minimizing the Temporal-Difference (TD) error which is defined as:

$$\mathcal{L}(\boldsymbol{w}, \boldsymbol{w}^-, \tau) = [Q_{\boldsymbol{w}}(s, a) - (r + \gamma \max_{a'} Q_{\boldsymbol{w}^-}(s', a'))]^2 \tag{1}$$

where $\boldsymbol{w}$ denotes the model weights, $\boldsymbol{w}^-$ is the weights of the target model, and $\tau = (s, a, r, s')$ is the transition sampled from the replay buffer $\mathcal{B}$.

Noisy Networks [22] introduce stochasticity into the model weights, reparameterizing them as:

$$\boldsymbol{w} = \boldsymbol{\mu} + \boldsymbol{\sigma} \cdot \tilde{\varepsilon} \tag{2}$$

where $\boldsymbol{\mu}$ and $\boldsymbol{\sigma}$ are the learnable weights, whereas $\tilde{\varepsilon}$ is a random vector sampled from an isotropic Gaussian distribution, adding randomness to the layer. (We denote element-wise product by '$\cdot$'.)

**Soft Actor-Critic.** Soft Actor-Critic (SAC) [24], a prevalent off-policy algorithm for continuous control, aims to maximize the expected return coupled with an entropy bonus. SAC consists of a policy $\pi$ and a critic model $Q$, each parameterized by weights $\boldsymbol{\theta}$ and $\boldsymbol{w}$ respectively.

In training, the critic $Q_{\boldsymbol{w}}$ is trained to minimize the following objective, defined as

$$\mathcal{L}_Q(\boldsymbol{\theta}, \boldsymbol{w}; \tau) = [Q_{\boldsymbol{w}}(s, a) - (r + \gamma(Q_{\boldsymbol{w}}(s', a') - \alpha \log \pi_{\boldsymbol{\theta}}(s', a')))]^2, \quad a' \sim \pi_{\boldsymbol{\theta}}(\cdot|s') \tag{3}$$

where $\alpha$ is the entropy coefficient and $\tau = (s, a, r, s')$ is the transition sampled from the buffer $\mathcal{B}$.

Subsequently, the policy $\pi_{\boldsymbol{\theta}}$ is jointly trained to maximize the following objective:

$$\mathcal{L}_\pi(\boldsymbol{\theta}, \boldsymbol{w}; s) = \mathbb{E}_{a \sim \pi_{\boldsymbol{\theta}}}[(Q_{\boldsymbol{w}}(s, a) - \alpha \log \pi_{\boldsymbol{\theta}}(a|s)]. \tag{4}$$

Throughout this paper, all of the plasticity-preserving methods (LN, SAM, Reset, CReLU) will be integrated and evaluated on top of these standard off-policy RL algorithms.

## 3 Synthetic Experiments

Analyzing the loss of plasticity in RL is indeed intricate given RL's multifaceted nature, encompassing challenges such as credit assignments, noisy targets, and the exploration-exploitation tradeoff. To alleviate this complexity, we design synthetic supervised learning experiments within a controlled framework using the CIFAR-10 dataset [37]. Our focus is to evaluate the model's adaptive capabilities under two distinct adaptation scenarios:

**Input Adaptation:** In RL, as the agent continually interacts with the environment, it constantly encounters new data through exploration. Effective adaptation to this data is paramount for effective decision-making. To simulate this scenario, we partitioned the training data into 100 chunks, sequentially adding them to a buffer. The training was conducted by sampling data from this progressively growing buffer. Overfitting to earlier data chunks and failing to adapt to newer ones would indicate performance degradation compared to a model trained on the entire dataset.

**Label Adaptation:** The RL domain often experiences shifts in the relationships between inputs and labels. Our synthetic experiment mirrored this dynamic by periodically altering labels during the training phase. The labels were randomly shuffled 100 times, with each class's labels uniformly reassigned, ensuring a consistent reassignment within a class (e.g., all 'cat' images transition from class 3 to class 4). A model's overfitting to initial relationships would impede its capability to adeptly adapt to evolving relationships.

Our experimental setup resonates with common design choices in RL [50, 26, 73], comprising three convolutional layers for the backbone and three fully connected layers for the head. Employing Stochastic Gradient Descent (SGD) with momentum [56, 54] as the optimizer, and a batch size of 128, the model underwent 50,000 updates, with 500 updates at each alternation step (i.e., appending a chunk for input adaptation and alternating labels for label adaptation). An exhaustive sweep was carried out for the learning rate and weight decay range from $\{0.1, 0.01, 0.001, 0.0001, 0.00001\}$, integrating weight decay to minimize variance across individual runs. Each methodology was trained over 30 random seeds, with results presented within a 95% confidence interval.

To delve into the impact of pursuing a smooth loss surface and preserving gradient propagation, we selected four distinct methods: Layer Normalization (LN) and Sharpness-Aware Minimization

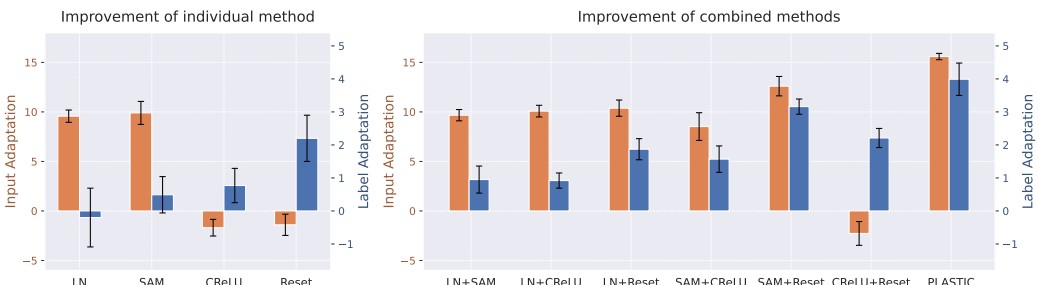

Figure 3: Comparative performance improvements of various methods and their combinations on input and label adaptation tasks. The left panel showcases individual method performance, while the right panel delves into the synergistic benefits of combining multiple methods.

(SAM) optimizer for the former, and Concatenated Rectified Linear Unit (CReLU) activation and Reset for the latter. For the SAM optimizer, the perturbation parameter was tuned across the set $\{0.1, 0.03, 0.01\}$, and for the Reset method, the reset interval was tuned over $\{5, 10, 20\}$ data chunks. Our analysis extended to exploring the synergistic interplay between these methods. To this end, we examined all unique pair-wise combinations of the selected methods (LN + SAM, ..., CReLU + Reset) with a combination of all methods. This exploration aimed to unveil synergetic effects between different methods in promoting smoother model landscapes and enhancing gradient propagation.

In Figure 3, we observed a clear distinction between the effects of various methods on input and label adaptation capabilities. The left panel illustrates the performance of applying individual methods. We found that LN and SAM excel in enhancing input adaptation, yet only offer marginal improvements for label adaptation. In contrast, CReLU and Reset yield modest enhancements in input adaptation while exhibiting a significant impact on label adaptation.

Turning our attention to the right panel, the focus is on the combined synergies of methods. The pairing of LN with SAM predominantly enhances input adaptation. When LN is merged with either CReLU or Reset, there's a marked improvement across both adaptation types. Likewise, the combination of SAM with either CReLU or Reset benefits both input and label adaptation scenarios. Among these combinations, the CReLU and Reset pairing is skewed towards improving label adaptation. Notably, PLASTIC, which integrates all of these methods yielded the most significant enhancements.

In summary, while individual methods are tailored towards specific enhancement areas, the integration of these techniques can yield synergies across both adaptation landscapes. These findings reinforce our notion of distinct yet complementary roles of seeking smooth loss surface and enhancing gradient propagation to improve the model's plasticity.

## 4 Experiments

In this section, we evaluate the effectiveness of enhancing the model's plasticity towards achieving sample-efficient reinforcement learning. Our experiments span two widely recognized benchmarks: the Atari-100k [9] for discrete control tasks and the DeepMind Control Suite (DMC) [65] for continuous control tasks. We have organized our experimental pursuits into four main subsections:

- Assessing the performance of the PLASTIC algorithm across both domains (Section 4.2).
- Analyzing the scaling behavior of PLASTIC. Specifically, we focus on the model's responsiveness with respect to the number of updates per environment interaction (Section 4.3).
- Exploring the advantages of improving plasticity on a large pre-trained model (Section 4.4).
- Ablation study for different combinations of plasticity-preserving methods (Section 4.5).

### 4.1 Experimental Setup

**Atari-100k.** Following the standard evaluation protocol from [32, 58], we evaluate the performance of 26 games on Arcade Learning Environments [73], limited to 100k interactions. The results are

measured by averaging over 10 independent trials. We use Data-Regularized Q-learning (DrQ) [73] as a base algorithm of PLASTIC, which is built on top of the Rainbow algorithm [26] and utilize data augmentation techniques to alleviate overfitting.

**Deepmind Control Suite Medium (DMC-M).**   To evaluate performance in the DMC benchmark, we adopt the evaluation protocol and network architecture identical to Nikishin et al. [52]. This benchmark consists of 19 continuous control tasks (i.e., 8 easy tasks and 11 medium tasks), where the agent controls its behavior from raw pixels. We selected 11 medium tasks and trained the agent for 2 million environment steps. Within this setting, we use Data-Regularized Q-learning (DrQ) algorithm [73], which incorporates data augmentation techniques in conjunction with the soft actor-critic algorithm [24]. The results are averaged over 10 random seeds.

**Baselines.**   This section presents a selection of plasticity-preserving methods designed to seek smooth loss surfaces or improve gradient propagation in RL. To seek a smooth loss surface, we consider Layer Normalization (LN) [47] and Sharpness-Aware Minimization Optimizer (SAM). For LN, we apply layer normalization after each convolutional layer in the backbone network. For SAM, we vary the perturbation parameter $\rho$ from $\{0.03, 0.1, 0.3\}$, selecting the model with the best performance.

For enhancing gradient propagation, we consider the re-initialization of the part of the networks (Reset) [52] and Concatenated ReLU activations (CReLU) [2]. Reset involves periodically resetting the parameters of the head layers to their initial distribution. Following [52], we reinitialize the parameters of the head layers' for every 40,000 gradient updates in Atari and 100,000 gradient updates in DMC. For CReLU, we replace ReLU activation with concatenated ReLU, which preserves both positive and negative features.

**Metrics.**   To gain a deeper understanding of the underlying geometry of the loss landscape (i.e., smoothness), we employ the maximum eigenvalue of the Hessian ($\lambda_{\max}$) [33]. This metric provides insights into the curvature of the loss function, where a larger value indicates a sharper and more intricate optimization landscape and a smaller value corresponds to a smoother and potentially more favorable landscape for optimization. To quantify the gradient propagation, we periodically record the fraction of the active units in the feature map by processing 512 transitions [47, 61, 18], randomly sampled from the replay buffer. When using ReLU activation, values below zero become inactive, which does not contribute to the updates of incoming weights.

To evaluate the performance of the agent, we report a bootstrapped interval for the interquartile mean (IQM), median, mean, and optimality gap, following the guidelines from [3]. For each environment, we average the performance of the trajectories at the end of training, 100 for Atari and 10 for DMC. For Atari, we normalize these scores to a Human Normalized Score (HNS) as HNS= $\frac{\text{agent\_score - random\_score}}{\text{human\_score - random\_score}}$, which quantifies the relative performance of the agent compared to human-level performance. More details of the experimental setup can be found in the Appendix.

## 4.2   Main Experiments

In this section, we present the experimental results aimed at evaluating the sample efficiency of the PLASTIC algorithm. Our examination involved contrasting its efficacy against individual plasticity-preserving methods. For this purpose, we used the Atari-100k and DMC-Medium benchmarks.

Table 1: **Performance on Atari-100k and DMC-M.** The results are averaged over 10 random seeds.

| Method | Atari-100k | | | DMC-M | | |
|---|---|---|---|---|---|---|
| | IQM | Median | Mean | IQM | Median | Mean |
| Base | 0.258 (0.224, 0.292) | 0.277 (0.209, 0.295) | 0.476 (0.432, 0.520) | 213 (146, 304) | 288 (223, 356) | 293 (239, 347) |
| SAM [21] | 0.325 (0.296, 0.354) | 0.327 (0.284, 0.368) | 0.501 (0.465, 0.537) | 278 (196, 371) | 341 (264, 404) | 332 (277, 389) |
| LN [47] | 0.259 (0.235, 0.293) | 0.247 (0.218, 0.276) | 0.463 (0.403, 0.522) | 412 (332, 488) | 415 (351, 491) | 421 (366, 477) |
| CReLU [2] | 0.256 (0.224, 0.287) | 0.193 (0.176, 0.252) | 0.498 (0.444, 0.553) | 338 (133, 566) | 398 (235, 549) | 394 (267, 522) |
| Reset [52] | 0.343 (0.314, 0.373) | 0.291 (0.231, 0.369) | 0.660 (0.611, 0.715) | 514 (434, 590) | 491 (430, 568) | 498 (442, 555) |
| PLASTIC | **0.421** (0.388, 0.457) | **0.347** (0.266, 0.422) | **0.933** (0.812, 1.067) | **565** (498, 626) | **540** (476, 599) | **537** (487, 587) |

Table 1 showcases the comparative outcomes of these methods. From the results, we observed that while individual plasticity-preserving methods enhanced sample efficiency across both discrete and continuous control benchmarks, the PLASTIC algorithm outperformed them all. This integration of multiple methods within PLASTIC clearly demonstrates superior results over any single approach. A comprehensive set of results for each environment is available in the Appendix.

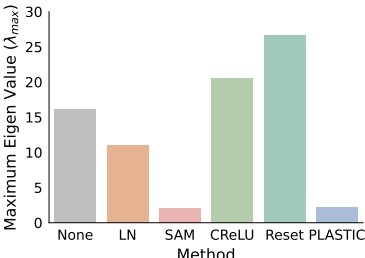 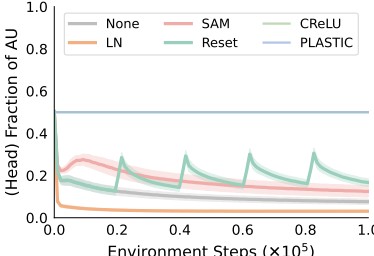

Figure 4: **Left:** The maximum eigenvalue of Hessian ($\lambda_{\max}(\nabla^2 \mathcal{L})$), representing the curvature of the loss landscape. **Right:** Fraction of active units in the head layers. Both metrics were derived from the DrQ algorithm's evaluation on the Atari-100k benchmark.

To delve deeper into the mechanisms driving these performance enhancements, we focused on two specific metrics: the maximum eigenvalue of the Hessian ($\lambda_{\max}$) and the fraction of active units in the head layers. As illustrated in Figure 4, methods such as LN and SAM encourage the model to converge on a smoother loss landscape. On the other hand, CReLU and Reset exhibit a pronounced presence of active units. Notably, by integrating these methods, PLASTIC succeeded in converging to a smooth loss region while maintaining a large number of active units.

Our prior synthetic experiments provided insight that convergence on a smoother loss landscape improves the model's input plasticity while enhanced gradient propagation enhances the model's label plasticity. Consequently, we conjecture that PLASTIC's superior performance is attributed to its capability to synergistically enhance both input and label plasticity.

### 4.3 Scaling Replay Ratio

Off-policy RL algorithms, in theory, promise enhanced sample efficiency by increasing the updates per environment interaction, commonly known as the replay ratio. Yet, a practical challenge arises: escalating updates often inversely affect sample efficiency due to the loss of the model's plasticity. This section delves into whether our PLASTIC algorithm can combat this decrement in plasticity, thereby improving sample efficiency with a scaling replay ratio.

For our analysis, we scaled the replay ratio up to 8 and evaluated the efficacy of PLASTIC on the Atari-100k benchmark. In addition, we compared PLASTIC against notable state-of-the-art (SOTA) model-based and model-free algorithms.

To ensure consistency and comparability in our evaluation, we computed the GPU hours for every method by leveraging their official codebases. For instance, PlayVirtual [76], MLR [78], SR-SPR [19], and PLASTIC can parallelize multiple runs on a single GPU. Thus, for a fair comparison, we followed the protocol as outlined by [19], dividing the total execution time by the total number of executed runs.

Given that SAM requires an auxiliary gradient computation during optimization, and CReLU increases the number of parameters, we introduced a simplified alternative, PLASTIC†. This simplified version only integrates Layer Normalization (LN) and the reset mechanism, ensuring minimal computational overhead. In addition, for each reset, we incorporate the Shrink & Perturb [4] method on the backbone network parameters. Shrink & Perturb softly reinitializes the model's parameters as $\theta_t = \alpha\theta_{t-1} + (1 - \alpha)\phi$, where $\phi \sim$ `initializer`. While Shrink & Perturb slightly lower performance at a replay ratio of 2, it demonstrated enhanced sample efficiency when scaling the replay ratio upwards.

As depicted in Table 2 and Figure 1, PLASTIC† establishes a Pareto frontier between Inter-Quartile Mean (IQM) and computational cost, illustrating its notable computational efficiency. While Ef-

Table 2: **Comparison to the SOTA on Atari-100k.** For IRIS, DreamerV3, EfficientZero, PlayVirtual, MLR, SR-SPR, and PLASTIC the results are averaged over 5, 5, 32, 15, 3, 10, and 5 seeds respectively.

| Type | Method | Search | Params (M) | RR | GPU hours | IQM | Median | Mean | OG |
|------|--------|--------|-----------|-----|-----------|-----|--------|------|-----|
| Model-Based | IRIS [49] | - | 30.4 | - | 36.3 | 0.501 | 0.289 | 1.046 | 0.512 |
| | DreamerV3 [25] | - | 17.9 | - | 5.1$^\dagger$ | 0.497 | 0.466 | 1.097 | 0.505 |
| | EfficientZero [75] | ✓ | 8.4 | - | 28.0 | n/a | 1.090 | 1.943 | n/a |
| Model-Free | PlayVirtual [76] | - | 7.4 | 2 | 4.9 | 0.374 | n/a | n/a | 0.558 |
| | MLR [78] | - | 161.7 | 2 | 3.7 | 0.432 | n/a | n/a | 0.522 |
| | SR-SPR [19] | - | 7.3 | 2 | 2.7 | 0.444 | 0.336 | 0.910 | 0.516 |
| | | | | 4 | 5.3 | 0.544 | 0.523 | 1.111 | 0.470 |
| | | | | 8 | 10.2 | 0.589 | 0.560 | 1.188 | 0.452 |
| | PLASTIC$^\dagger$ | - | 6.8 | 2 | 0.5 | 0.396 | 0.425 | 0.702 | 0.541 |
| | | | | 4 | 1.0 | 0.518 | 0.517 | 0.858 | 0.478 |
| | | | | 8 | 1.9 | 0.583 | 0.542 | 0.939 | 0.448 |
| | PLASTIC | - | 7.2 | 2 | 1.0 | 0.421 | 0.347 | 0.933 | 0.535 |
| | | | | 4 | 1.9 | 0.545 | 0.407 | 1.002 | 0.475 |
| | | | | 8 | 3.8 | 0.571 | 0.494 | 0.968 | 0.461 |

ficientZero exhibits state-of-the-art performance, it uniquely employs a search algorithm coupled with a domain-specific heuristic, namely early environment resets. When excluding these specific elements, PLASTIC stands out as a strong competitor against the other methods.

Through these results, we found that PLASTIC can effectively prevent the loss of plasticity by incorporating various plasticity-preserving strategies. Furthermore, for practitioners, the construction of the Pareto frontier by PLASTIC$^\dagger$ is particularly beneficial. By simply incorporating layer normalization and implementing resets in the underlying off-policy RL algorithms, they can achieve improved sample efficiency. This comes with the advantage of minimal computational overhead and only requires minor code adjustments.

In conclusion, our experiments underscore the potential of leveraging enhanced plasticity as a means to improve sample efficiency in RL.

## 4.4 Compatibility with a Large Pretrained Model

Recent years have seen growing interest in leveraging large pretrained models to improve sample efficiency in RL [59, 60, 7]. We investigate how combining PLASTIC's principles with these models can counteract the loss of plasticity, a common obstacle to rapid adaptation.

For our evaluation, we selected the SimTPR model [41]. Using a 30-layer convolutional network, SimTPR is pretrained via self-predictive representation learning on suboptimal Atari video datasets. As SimTPR does not employ Layer Normalization (LN) and Concatenated ReLU (CReLU) activations, we employ SAM to seek a smoother loss landscape and use Reset techniques to facilitate gradient propagation. Following the approach in section 4.3, for each reset, we applied Shrink & Perturb to the backbone network. In the fine-tuning phase, we initialized an MLP-based head network on top of the frozen, pretrained backbone network. This head network underwent training for 100k environment steps, leveraging the Rainbow algorithm. See the Appendix section for more details.

Table 3 provides a summary of applying either SAM or Reset to the pretrained model. Solely scaling the replay ratio during fine-tuning leads to a noticeable decrement in its efficacy. However, we observed that the usage of Reset or SAM can counteract this decrease, leading to a pronounced enhancement in performance. Furthermore, the integration of both techniques surpassed individual contributions, indicating the presence of synergy.

These observations imply that elevating both input and label plasticity can play a pivotal role in enhancing sample efficiency, even in large, pre-trained models.

Table 3: **Fine-tuning from a pretrained model.** Reset$^\dagger$ applies a soft reset to the backbone and a hard reset to the head.

| RR | SAM | Reset$^\dagger$ | IQM |
|-----|-----|-------|-----|
| 2 | | | 0.366 (0.324, 0.397) |
| | | ✓ | 0.709 (0.650, 0.745) |
| | ✓ | ✓ | 0.776 (0.703, 0.854) |
| 4 | | | 0.243 (0.214, 0.267) |
| | | ✓ | 0.780 (0.706, 0.865) |
| | ✓ | ✓ | 0.834 (0.769, 0.889) |

## 4.5 Ablation Studies

To further analyze the interplay between seeking a smooth loss surface and improving gradient propagation, we have conducted a series of ablation studies for different combinations of plasticity-preserving methods. Here, we averaged the performance over 10 random seeds.

Table 4 showcases the results when applying various combinations to the Atari-100k benchmark. The first five and the last rows echo the results from Table 1. It becomes clear that the concurrent pursuit of a smooth loss surface and the enhancement of gradient propagation surpasses the individual application of either strategy (for instance, LN + CReLU > LN ≈ CReLU). The comprehensive integration of all methods, referred to as PLASTIC, demonstrates the highest performance.

For practitioners considering our approach, it's worth highlighting its adaptability. The methodology does not necessitate the application of all methods across every plasticity category to reap substantial benefits. Instead, deploying just one technique from each category can yield notable improvements. For scenarios constrained by computational resources, using LN and Reset is recommended over SAM, due to the latter's extra computational burden, and CReLU, given its added parameters (as noted in PLASTIC[†] in Section 4.3). Additionally, when one is working with a pre-trained network where altering the overall architecture is not feasible, LN and CReLU might not be practical since they inherently change the network's structure. In such instances, as detailed in Section 4.4, the combination of SAM and Reset emerges as a potent solution to improve downstream performance.

Table 4: Performance comparison of various plasticity-preserving method combinations on the Atari-100k benchmark.

| LN | SAM | CReLU | Reset | IQM |
|----|-----|-------|-------|-----|
|    |     |       |       | 0.258 (0.224, 0.292) |
| ✓  |     |       |       | 0.259 (0.235, 0.293) |
|    | ✓   |       |       | 0.325 (0.296, 0.354) |
|    |     | ✓     |       | 0.256 (0.224, 0.287) |
|    |     |       | ✓     | 0.343 (0.314, 0.373) |
| ✓  | ✓   |       |       | 0.341 (0.325, 0.366) |
| ✓  |     | ✓     |       | 0.284 (0.257, 0.314) |
| ✓  |     |       | ✓     | 0.396 (0.365, 0.430) |
|    | ✓   | ✓     |       | 0.372 (0.357, 0.408) |
|    | ✓   |       | ✓     | 0.411 (0.377, 0.447) |
|    |     | ✓     | ✓     | 0.373 (0.344, 0.402) |
| ✓  | ✓   | ✓     | ✓     | **0.421 (0.388, 0.457)** |

## 5 Conclusion, Limitations, and Future Work

In this paper, we aimed to address the prevalent issue of the "loss of plasticity" in off-policy RL algorithms. Through synthetic experiments on CIFAR-10, we discovered that finding a smoother point of loss landscape largely improves input plasticity, while maintaining effective gradient propagation enhances label plasticity. From these insights, we proposed PLASTIC, an algorithm that synergistically combines the SAM optimizer, LN, periodic Reset, and CReLU activation. Empirically, this combination substantially improves both forms of plasticity. Demonstrating robust performance on benchmarks including Atari-100k, PLASTIC offers a promising avenue for advancing sample efficiency in RL.

Nevertheless, our study has certain constraints. Our empirical evaluations were primarily limited to the Atari and DMC environments. A compelling direction for future work would be to assess PLASTIC's efficacy in more intricate settings, such as MetaWorld [79], or Procgen [15]. These environments introduce greater non-stationarities, challenging the model's adaptability.

While the smoothness of loss surfaces and gradient propagation are fundamental to our findings, they might not capture the full complexity of the "loss of plasticity" phenomenon. Notably, even though SAM was sufficient to find the smoothest loss surfaces, Layer Normalization's integration amplifies its efficacy. Similarly, while CReLU inherently mitigates the reduction of active units, introducing periodic resets offers pronounced enhancements. We hypothesize that the nuanced attributes like the smoothness of loss landscapes and the number of active units might only scratch the surface of deeper determinants influencing model plasticity. Therefore, an in-depth understanding and measurement of network plasticity can pave the way for more foundational solutions.

In conclusion, while we acknowledge the constraints, our research offers practioncal insights for amplifying the plasticity of RL agents We hope our findings open up a diverse possibility for future exploration, potentially leading to more sample-efficient and adaptable algorithms in RL.

## Acknowledgments and Disclosure of Funding

This work was supported by the Institute of Information & Communications Technology Planning & Evaluation (IITP) grant funded by the Korean government (MSIT) (No.2019-0-00075, Artificial Intelligence Graduate School Program(KAIST)). HL, HK, and JC acknowledge support from the National Supercomputing Center with supercomputing resources including technical support (KSC-2023-CRE-0074). HC and CY acknowledge support from the Institute of Information & communications Technology Planning & Evaluation (IITP) grant (No. 2022-0-00184, Development and Study of AI Technologies to Inexpensively Conform to Evolving Policy on Ethics) and the National Research Foundation of Korea (NRF) grant (No. RS-2023-00211352), both funded by the Korea government (MSIT).

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

# Appendix

# A Implementation Details

## A.1 Computation

For all experiments, we used an NVIDIA RTX 3090 GPU for neural network training and a 32-core AMD EPYC 7502 for multi-threaded tasks, accelerating machine learning model training and inference.

For the Atari 100k benchmark, our computations were based on the DrQ algorithm with SAM (Sharpness-Aware Minimization) and Reset. We ran 3 experiments in parallel on a single GPU where PLASTIC and PLASTIC[†] took 3 and 1.5 hours, which effectively translates to an average of 1 and 0.5 hours per run.

For the DeepMind Control Suite (DMC), runtimes varied based on the individual environment due to the differences in action repeat (i.e., the number of the same actions taken for each environmental interaction). On our experimental setup, training took roughly 8 hours to complete 2 million environment steps with an action repeat of 4, and around 15 hours with an action repeat of 2.

## A.2 Synthetic Experiment

In our synthetic experiment, we utilized the CIFAR-10 dataset [38] to test both input adaptation and label adaptation scenarios. The dataset comprises 50,000 training images and 10,000 test images, evenly distributed across 10 classes. Each image in the dataset is a color image of size 32x32.

**Input Adaptation:** In the input adaptation experiment, the training data was segmented into 100 chunks, each consisting of 500 images. These chunks were sequentially added to a buffer, which progressively expanded during the course of the experiment. For every model update, data were randomly sampled from this progressively growing buffer.

**Label Adaptation:** In the label adaptation experiment, label alterations were done 100 times during the training phase. The process of label reassignment was performed in a uniform manner across all classes. For example, if the original class labels ranged from 0 to 9, after each label alteration, all data points within a specific class would be consistently reassigned to a new class within the same range.

We selected a model architecture that reflects common architectural designs utilized in reinforcement learning research [50, 26]. The architecture consists of three convolutional layers for the backbone and three fully-connected layers for the head. The three convolutional layers employed kernel sizes of 3x3, with strides of (3, 3, 1), and the number of channels was set as (32, 64, 64) respectively. Subsequent to this, the three fully-connected layers consisted of hidden dimensions of (512, 128, 10).

The model was trained for a total of 50,000 updates, with a batch size of 128 for each update (i.e., equivalent to training 128 epochs). The training was conducted using Stochastic Gradient Descent (SGD) with momentum. To select the learning rates and weight decay rates, we vary the values range from $\{0.1, 0.01, 0.001, 0.0001, 0.00001\}$, integrating weight decay to minimize variance across individual runs.

In the synthetic experiment, we utilized four distinct methods: Layer Normalization (LN), Sharpness-Aware Minimization, Concatenated Rectified Linear Unit (CReLU), and Reset. For the SAM optimizer, we vary the perturbation parameter across the set $\{0.1, 0.03, 0.01\}$ and for the Reset method, we search the reset interval over $\{5, 10, 20\}$ data chunks.

Finally, we reported the test accuracy, averaged over 30 random seeds with a 95% confidence interval, at the end of each experiment.

## A.3 Arcade Learning Environment (Atari-100k)

We implemented the Rainbow algorithm [26] while following design choices from DER [67] and DrQ [73]. Furthermore, we integrated modifications suggested by D'Oro et al [19], including the application of a momentum encoder for the target network and the use of this target network for exploration. While these adjustments had little impact under a low replay ratio regime (<=2), they played a significant role when scaling the replay ratio with the reset mechanism [52]. A detailed hyperparameter is described in Table 5.

In the case of LN, we apply layer normalization after each convolutional and fully-connected layer. For CReLU, we replaced all ReLU activations with concatenated ReLU where the input dimensions are doubled for each subsequent layers. For the methods using the Reset mechanism, we reset the head layer's parameters every 40,000 gradient updates, in accordance with the original paper [52].

Table 5: **Hyperparameters on Atari 100k.** The ones introduced by this work are at the bottom.

| Hyperparameter | Value |
|---|---|
| State downsample size | (84, 84) |
| Grey scaling | True |
| Data augmentation | Random Shifts and Intensity Jittering |
| Random Shifts | $\pm 4$ pixels |
| Intensity Jiterring scale | 0.05 |
| Frame skip | 4 |
| Stacked frames | 4 |
| Action repeat | 4 |
| Training steps | 100k |
| Update | Distributional Q |
| Dueling | True |
| Support of Q-distribution | 51 |
| Discount factor $\gamma$ | 0.99 |
| Batch size | 32 |
| Optimizer $(\beta_1, \beta_2, \epsilon)$ | Adam (0.9, 0.999, 0.000015) |
| Learning rate | 0.0001 |
| Max gradient norm | 10 |
| Priority exponent | 0.5 |
| Priority correction | $0.4 \rightarrow 1$ |
| EMA coefficient $(\tau)$ | 0.99 |
| Exploration Network | Target |
| Exploration | Noisy nets |
| Noisy nets parameter | 0.5 |
| Replay buffer size | 100k |
| Min buffer size for sampling | 2000 |
| Replay ratio | 2 |
| Multi-step return length | 10 |
| Q-head hidden units | 512 |
| Q-head non-linearity | ReLU |
| Evaluation trajectories | 100 |
| SAM parameter $\rho$ | 0.1 |
| Reset interval | $40,000$ |

## A.4 Deepmind Control Suite Medium (DMC-M)

We utilized an open-source JAX implementation provided by Nikishin et al. [52] as the foundation for our work. We integrated the SAM optimizer implementation into this existing framework. To ensure consistency and comparability with prior studies, we strictly followed the architecture and hyperparameters outlined by Nikishin et al. [52], documented in Table 6.

In the case of the Base, L2, and Reset models described in Table 6, we adopted the results from Nikishin et al [52]. For the LN model, we incorporated layer normalization after each convolutional and fully-connected layer. Additionally, we replaced all ReLU activations with concatenated ReLU for the CReLU baseline.

We utilized the same training steps of 2,000,000 and reset interval of 100,000 for the DMC medium task as used by Nikishin et al [52].

Table 6: **Hyperparameters on DMC.** The ones introduced by this work are at the bottom.

| Parameter | Setting |
|---|---|
| Gray-scaling | True |
| Observation down-sampling | (64, 64) |
| Frame stacked | 3 |
| Discount factor | 0.99 |
| Minibatch size | 512 |
| Learning rate | 0.0003 |
| Backbone: channels | 32, 64, 128, 256 |
| Backbone: stride | 2, 2, 2, 2 |
| Backbone: latent dim | 50 |
| Head: n. hidden layers | 2 |
| Head: hidden units | 256 |
| Target network update period | 1 |
| EMA coefficient $\tau$ | 0.995 |
| Initial Temperature | 0.1 |
| Updates per step | 1 |
| Replay Buffer Size | $1,000,000$ |
| Total training steps | $2,000,000$ |
| Evaluation trajectories | 100 |
| SAM parameter $\rho$ | Quadruped : 0.1
Others : 0.01 |
| Reset interval | $100,000$ |

# B Applying SAM to Deep RL

## B.1 Preliminaries and Backgrounds on SAM

The capability to make a correct prediction when given unseen inputs, which we refer to as *input plasticity*, has been recognized as an important issue across the entirety of machine learning research, and numerous studies have been conducted to address it. Keskar et al. [34] propose a sharpness measure, arguing that finding wide and smooth minima is beneficial in terms of generalizability (i.e., performance on unseen datasets). Stochastic Weight Averaging (SWA) finds a flat minimum by averaging checkpoints in the model's training trajectory [30]. The work by [71] incorporates the addition of adversarial weight perturbation over the course of training. These works are based on the hypothesis by [34] that wide and flat minima result in a more plastic model under change of inputs.

Following a similar philosophy, Foret et al. [21] devised sharpness-aware minimization (SAM), which aims to reduce not only the value of the training loss but also its sharpness. By reducing the sharpness of the loss and encouraging the model to get closer to a flat minimum, SAM has been successfully applied in various settings including computer vision [14], natural language processing [6], meta-learning [1], and model compression [51].

The aforementioned studies concern training in stationary data distributions. In contrast, our work involves experiments in reinforcement learning which has non-stationary data distribution in which the efficacy of SAM has not yet been verified. Indeed, we empirically verified that the usage of SAM enhances the adaptability of deep RL agents.

Now, we elaborate on what SAM is. Originally, SAM aims to train a model which is robust to adversarially perturbed model weights, by solving the following bi-level optimization problem on the loss function $\mathcal{L}$:

$$\min_{\boldsymbol{w}} \left\{ \max_{\boldsymbol{\epsilon}:\|\boldsymbol{\epsilon}\|_2 \leq \rho} \mathcal{L}(\boldsymbol{w} + \boldsymbol{\epsilon}) \right\}. \tag{5}$$

This problem implies minimization of loss value around an $\ell_2$-ball of radius $\rho$, which encourages a model parameter $\boldsymbol{w}$ to find a smoother region of the loss landscape. Here, $\rho > 0$ is a hyperparameter, called *SAM parameter*, restricting the size of perturbation to make the bi-level problem (5) feasible. Since the exact solution of the inner maximization problem in (5) is not tractable yet, first-order Taylor approximation is applied as a remedy for finding an approximately optimal perturbation at $\boldsymbol{w}$,

$$\boldsymbol{\epsilon}_\rho^*(\boldsymbol{w}) := \frac{\rho}{\|\nabla_{\boldsymbol{w}} \mathcal{L}(\boldsymbol{w})\|_2} \nabla_{\boldsymbol{w}} \mathcal{L}(\boldsymbol{w}).$$

We call the step of computing $\boldsymbol{\epsilon}_\rho^*(\boldsymbol{w})$ as the *perturbation step* of SAM. As a result of the perturbation step, we can approximately solve the problem (5) with any gradient update algorithm (*i.e.*, *base optimizer*), such as Stochastic Gradient Descent (SGD) [56], SGD with momentum [54], or Adam [35], using the *SAM gradient* at current weight $\boldsymbol{w}_t$: $\nabla_{\boldsymbol{w}} \mathcal{L}(\boldsymbol{w}_t + \boldsymbol{\epsilon}_\rho^*(\boldsymbol{w}_t))$. We call the step of updating the weights with SAM gradient as the *update step* of SAM. In essence, SAM is an iterative algorithm alternating perturbation and update steps. Readers can check a more detailed derivation of the SAM gradient in [21].

It is both theoretically and empirically proven that such an optimization scheme implicitly prefers the weights in smoother (i.e., less *sharp*) region of loss landscape, in terms of, *e.g.*, the maximum eigenvalue and/or trace of Hessian [21, 31, 70, 62].

Naturally, in (deep) RL, we never have access to the full static loss function as described above. Hence, analogous to most of the applications of SAM, it is natural to randomly sample a mini-batch $\mathcal{B}_t$ of $m$ transitions from replay buffer and apply SAM update (often called $m$-SAM) using the stochastic gradient $\nabla_{\boldsymbol{w}} \mathcal{L}_{\mathcal{B}_t}\left(\boldsymbol{w}_t + \boldsymbol{\epsilon}_{\rho,\mathcal{B}_t}^*(\boldsymbol{w}_t)\right)$, where

$$\boldsymbol{\epsilon}_{\rho,\mathcal{B}}^*(\boldsymbol{w}) := \frac{\rho}{\|\nabla_{\boldsymbol{w}} \mathcal{L}_{\mathcal{B}}(\boldsymbol{w})\|_2} \nabla_{\boldsymbol{w}} \mathcal{L}_{\mathcal{B}}(\boldsymbol{w}).$$

There are several considerations for applying SAM to RL agents. When applying SAM to the Rainbow agent, the random noises from noisy layers embedded in the agent might hurt the SAM perturbation. However, we observe that the effect of regulating the noises is not significant. Moreover, there are options to apply SAM perturbation solely to the backbone or head. We find that applying SAM to the whole network is the most beneficial for enhancing generalization capability, yet we

observe that solely applying SAM to the backbone is quite similar to the case of applying it to the whole. On the other hand, when applying SAM to the SAC agent, there are multiple loss functions to be optimized, which share parameters. Again, applying SAM to both actor and critic is the most desirable option, yet we observe that the application to critic is more critical for performance improvement. More detailed ablation studies appear in Section B.4.

## B.2 Deep Q-Learning with SAM and Noisy Layers

We used the Rainbow agent and its variants to learn discrete control problems of the Atari-100k benchmark. Although there are many other components inside the Rainbow agent, such as multi-step learning, distributional RL, and dueling Deep Q-Network (DQN), it would be confusing to integrate all details together in the pseudocode. Thus, for simplicity of the display, we provide a pseudocode of applying SAM to vanilla DQN with noisy layers. See Algorithm 1.

---

**Algorithm 1:** Deep Q-Learning with Noisy Layers & SAM

1 **Input:** Learning rate $\eta$; SAM parameter $\rho$; Replay ratio $R$; Discount factor $\gamma$; Polyak averaging factor $\tau$ ;

2 **Initialize:** Replay memory $\mathcal{D}$; Q-function weight $\boldsymbol{w}$; Target weight $\boldsymbol{w}^- \leftarrow \boldsymbol{w}$ ;

3 **foreach** environment step $t = 1, 2, \ldots$ **do**

4      Collect a trajectory $(s, a, r, s', d)$ with $\varepsilon$-greedy   $(d = \mathbb{1}[s'$ is terminal$])$;

5      Store the trajectory to $\mathcal{D}$ ;

6      If $d = 1$, Reset environment state;

7      **foreach** optimization step $i \in \{1, \ldots, R\}$ **do**

8          Sample a minibatch of transitions $\mathcal{B} = \{(s, a, r, s', d)\}$ from $\mathcal{D}$ ;

9          Compute TD target $y(r, s', d) = r + \gamma(1 - d) \max_{a'} Q_{\boldsymbol{w}^-}(s', a')$;

         // Perturbation step

10          Sample a random noise $\boldsymbol{\xi}$ of noisy layers inside Q-function;   $\boldsymbol{w} \leftarrow \boldsymbol{w} + \boldsymbol{\xi}$;

11          Compute gradient $\boldsymbol{g}(\boldsymbol{w}) = \nabla_{\boldsymbol{w}} \dfrac{1}{|\mathcal{B}|} \displaystyle\sum_{(s,a,r,s',d) \in \mathcal{B}} (y(r, s', d) - Q_{\boldsymbol{w}}(s, a))^2$ ;

12          Compute SAM perturbation $\tilde{\boldsymbol{w}}^{\mathrm{SAM}} = \boldsymbol{w} + \dfrac{\rho}{\|\boldsymbol{g}(\boldsymbol{w})\|_2} \boldsymbol{g}(\boldsymbol{w})$ ;

         // Update step

13          Sample a random noise $\boldsymbol{\xi}'$ of noisy layers inside Q-function;   $\tilde{\boldsymbol{w}}^{\mathrm{SAM}} \leftarrow \tilde{\boldsymbol{w}}^{\mathrm{SAM}} + \boldsymbol{\xi}'$;

14          Compute SAM gradient $\boldsymbol{g}^{\mathrm{SAM}} = \boldsymbol{g}(\tilde{\boldsymbol{w}}^{\mathrm{SAM}})$;

15          Gradient descent update $\boldsymbol{w} \leftarrow \boldsymbol{w} - \eta \boldsymbol{g}^{\mathrm{SAM}}$; // Could be modified as any gradient-based optimizer

16          Update target weight $\boldsymbol{w}^- \leftarrow \tau \boldsymbol{w}^- + (1 - \tau)\boldsymbol{w}$ ;

---

### B.3 Soft Actor-Critic with SAM

We also provide a pseudocode for applying SAM to the SAC algorithm. See Algorithm 2.

---

**Algorithm 2:** Soft Actor-Critic with SAM

---

1 **Input:** Learning rates $\eta_Q, \eta_\pi, \eta_\alpha$; SAM parameters $\rho_Q, \rho_\pi, \rho_\alpha$; Replay ratio $R$; discount factor $\gamma$; Polyak averaging factor $\tau$; minimum expected entropy $\mathcal{H}$;

2 **Initialize:** Replay memory $\mathcal{D}$; Q-function weights $\boldsymbol{w}_1, \boldsymbol{w}_2$; Q-function weights $\boldsymbol{w}_1^-, \boldsymbol{w}_2^-$; Policy weight $\boldsymbol{\theta}$; Entropy regularization coefficient $\alpha$ ;

3 **foreach** environment step $t = 1, 2, \ldots$ **do**

4      Observe state $s$ and sample/execute an action $a \sim \pi_{\boldsymbol{\theta}}(\cdot|s)$;

5      Observe reward $r$, next state $s'$, and done signal $d = \mathbb{1}[s'$ is terminal];

6      Store the trajectory $(s, a, r, s', d)$ to $\mathcal{D}$ ;

7      If $d = 1$, Reset environment state;

8      **foreach** optimization step $i \in \{1, \ldots, R\}$ **do**

9          Sample a minibatch of transitions $\mathcal{B} = \{(s, a, r, s', d)\}$ from $\mathcal{D}$ ;

         // Perturbation step

10          Compute target

$$y(r, s', d; \boldsymbol{\theta}) = r + \gamma(1 - d)\left(\min_{i=1,2} Q_{\boldsymbol{w}_i^-}(s', a') - \alpha \log \pi_{\boldsymbol{\theta}}(a'|s')\right) \quad (a' \sim \pi_{\boldsymbol{\theta}}(\cdot|s));$$

11          Compute Q-function gradient:

$$\boldsymbol{g}_{Q,i}(\boldsymbol{w}_i) = \nabla_{\boldsymbol{w}_i} \frac{1}{|\mathcal{B}|} \sum_{(s,a,r,s',d)\in\mathcal{B}} \left(y(r, s', d; \boldsymbol{\theta}) - Q_{\boldsymbol{w}_i}(s, a)\right)^2; \quad (i = 1, 2)$$

12          Compute policy gradient by sampling differentiable action $\tilde{a}_{\boldsymbol{\theta}}(s) \sim \pi_{\boldsymbol{\theta}}(\cdot|s)$:

$$\boldsymbol{g}_\pi(\boldsymbol{w}) = \nabla_{\boldsymbol{\theta}} \frac{1}{|\mathcal{B}|} \sum_{s\in\mathcal{B}} \left(\min_{i=1,2} Q_{\boldsymbol{w}_i}(s, \tilde{a}_{\boldsymbol{\theta}}(s)) - \alpha \log \pi_{\boldsymbol{\theta}}(\tilde{a}_{\boldsymbol{\theta}}(s)|s)\right)^2;$$

13          Compute temperature gradient: $\boldsymbol{g}_{\text{temp}}(\alpha) = \nabla_\alpha \frac{1}{|\mathcal{B}|} \sum_{s\in\mathcal{B}} \left(-\alpha \log \pi_{\boldsymbol{\theta}}(\tilde{a}_{\boldsymbol{\theta}}(s)|s) - \alpha\mathcal{H}\right)$;

14          SAM-perturbations:

$$\tilde{\boldsymbol{w}}_i^{\text{SAM}} = \boldsymbol{w}_i + \frac{\rho_Q}{\|\boldsymbol{g}_{Q,i}(\boldsymbol{w}_i)\|_2} \boldsymbol{g}_{Q,i}(\boldsymbol{w}_i); \quad (i = 1, 2)$$

$$\tilde{\boldsymbol{\theta}}^{\text{SAM}} = \boldsymbol{\theta} + \frac{\rho_\pi}{\|\boldsymbol{g}_\pi(\boldsymbol{\theta})\|_2} \boldsymbol{g}_\pi(\boldsymbol{\theta});$$

$$\tilde{\alpha}^{\text{SAM}} = \alpha + \frac{\rho_\alpha}{\|\boldsymbol{g}_{\text{temp}}(\alpha)\|_2} \boldsymbol{g}_{\text{temp}}(\alpha);$$

         // Update step

15          Compute target $y(r, s', d; \tilde{\boldsymbol{\theta}}^{\text{SAM}})$;

16          Compute SAM gradients:

$$\boldsymbol{g}_{Q,i}^{\text{SAM}} = \boldsymbol{g}_{Q,i}(\tilde{\boldsymbol{w}}_i^{\text{SAM}}), \;\; \boldsymbol{g}_\pi^{\text{SAM}} = \boldsymbol{g}_\pi(\tilde{\boldsymbol{\theta}}^{\text{SAM}}), \;\; \boldsymbol{g}_{\text{temp}}^{\text{SAM}} = \boldsymbol{g}_{\text{temp}}(\tilde{\alpha}^{\text{SAM}});$$

17          Gradient descent updates:

$$\boldsymbol{w}_i \leftarrow \boldsymbol{w}_i - \eta_Q \boldsymbol{g}_{Q,i}^{\text{SAM}}; \;\; \boldsymbol{\theta} \leftarrow \boldsymbol{\theta} - \eta_\pi \boldsymbol{g}_\pi^{\text{SAM}}; \;\; \alpha \leftarrow \alpha - \eta_\alpha \boldsymbol{g}_{\text{temp}}^{\text{SAM}};$$

         // Could be modified as any gradient-based optimizer

18          Update target weight $\boldsymbol{w}_i^- \leftarrow \tau \boldsymbol{w}_i^- + (1 - \tau)\boldsymbol{w}_i$ ;

---

## B.4 Ablations on the usage of SAM

We conduct some ablation studies aiming to answer the following questions:

1. How should we deal with *noisy layers* during training?
2. What if we solely apply SAM perturbation to the *backbone* or *head* of Rainbow agent?
3. What if we solely apply SAM perturbation to the *actor* or *critic* of SAC agent?

The first two questions are answered by experiments with the DrQ algorithm tested on the Atari-100k benchmark, whereas for the last question we test on the DMC-M benchmark with the SAC agent.

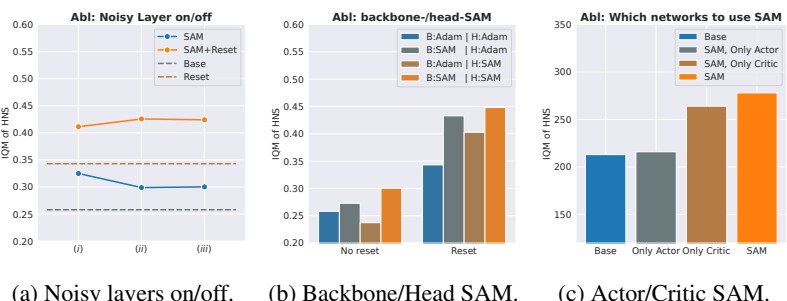

(a) Noisy layers on/off.    (b) Backbone/Head SAM.    (c) Actor/Critic SAM.

Figure 5: **Ablation study on utilizing SAM**.

**Noisy Layers.** For the case of the Rainbow agent, we compare three different schemes of SAM updates regarding random noises from noisy layers: (i) perturbation with independent noise, using independently sampled (*e.g.*, different) noise between perturbation and update step (default setting); (ii) update with reused noise, turning the noise on at the perturbation step and reusing that noise at the update step; (iii) noise-less perturbation, turning the noise off at the perturbation step and turning it on at the update step. Despite the variations, the overall impact on the agent's performance was relatively consistent. Note that option (i) is the most naïve approach which can be implemented by simply adding a SAM optimizer code block into any RL implementation with noisy layers.

**Backbone-SAM v.s. Head-SAM.** We experiment with resectioning the parts to be updated with SAM in the RL agent architecture. We temporarily turn off the gradient of either backbone or head at the perturbation step and then turn on that gradient back at the update step. The result of the experiment, presented in Figure 5b, says that updating with backbone-only perturbation is more beneficial than using head-only perturbation, although applying both is much better. Indeed, SAM perturbation of the head alone hurts the performance without any other SAM perturbation or resetting.

We want to explain why these happen from the point of view of the number of active neurons in the backbone/head. Interestingly, we observe that the SAM perturbation of backbone parameters induces *more* active units in the head, whereas the SAM perturbation of head parameters results in *less* active units in the backbone. Recall that both tendencies are observed when we apply SAM perturbation to both the backbone and head at once. From this, we can say that the rise of active units in the head by applying SAM is due to the SAM perturbation of the backbone, whereas the reason for the backbone getting sparser is the SAM perturbation of the head.

**Actor-SAM v.s. Critic-SAM.** SAC is an actor-critic algorithm that employs three distinct sets of updating parameters, namely the actor, critic, and alpha. Since alpha has just one parameter, we focus on investigating the impact of SAM on actor and critic parameters. We conduct an ablation study utilizing the identical configuration as the DMC experiment presented in Section 4.2. Figure 5c demonstrates that exclusively applying SAM to critic parameters is more critical than solely applying SAM to actor parameters while using both is much better.

In the context of SAC, the actor and critic networks are constructed to jointly utilize the backbone layer. In order to ensure stable learning, updates to the backbone layer are exclusively driven by critic losses, while actor losses do not influence the backbone layer [73, 72]. Consequently, if SAM is exclusively applied to the actor parameters, the backbone layer remains unaffected by SAM as

there is no gradient propagation. As a result, the performance of the SAM solely used on the actor parameters, which has no impact on the backbone, is inferior to other approaches incorporating SAM, which corroborates the aforementioned findings.

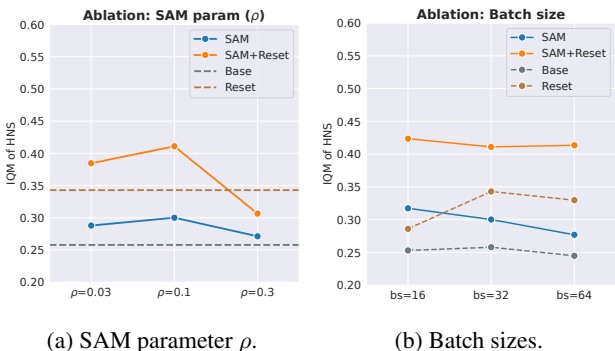

(a) SAM parameter $\rho$.    (b) Batch sizes.

Figure 6: More ablation studies on using SAM.

**SAM parameter ($\rho$).** Being the main hyperparameter of SAM, the $\rho > 0$ represents the intensity of SAM perturbation. In every other experiment with DER and DrQ(-Rainbow), we do not tune the value of $\rho$ but fix them as 0.1. Here, however, we run two other values of $\rho$'s, 0.03 and 0.3, to examine the effect of $\rho$. The result is shown in Figure 6a. We definitely observe the trade-off due to the size of $\rho$. If the $\rho$ is too small, the scale of SAM perturbation becomes negligible and it cannot drastically improve the generalization capability compared to the case without SAM. However, a larger value of $\rho$ is not always beneficial; the generalization capability does not always scale with the size of SAM perturbation. We, however, claim that finding an appropriate value of $\rho$ is not too difficult; our recommendation is to test a few values between 0.01 and 1.

**Batch size.** We further study robustness with respect to batch sizes. We investigate the batch sizes among 16, 32 (baseline), and 64. From the resulting plot Figure 6b, we observe that SAM consistently improves the performance regardless of the choice of batch size, by comparing solid lines (SAM+Adam) and dashed lines (Adam). We also observe that the performance improvement between reset and non-reset becomes much larger when SAM is applied.

# C Per Environment Results

## C.1 Atari-100k

## C.2 DMC-M

Table 7: **Mean trajectory scores on DrQ algorithm.** We report the mean trajectory scores on the 26 Atari games, evaluated on top of DrQ. For each random seed, the results are averaged over 100 different trajectories at the end of training. The results are then averaged over 10 random seeds.

| Game | Random | Human | Base | SAM | LayerNorm | CReLU | Reset | PLASTIC |
|---|---|---|---|---|---|---|---|---|
| Alien | 227.8 | 7127.7 | 757.62 | 854.84 | 899.14 | 1011.84 | 926.73 | 1032.0 |
| Amidar | 5.8 | 1719.5 | 164.12 | 187.76 | 176.12 | 146.69 | 158.22 | 201.6 |
| Assault | 222.4 | 742.0 | 582.59 | 597.9 | 566.26 | 585.57 | 694.17 | 888.5 |
| Asterix | 210.0 | 8503.3 | 823.3 | 780.6 | 889.3 | 887.8 | 921.15 | 1066.0 |
| BankHeist | 14.2 | 753.1 | 48.36 | 185.25 | 56.16 | 79.88 | 141.14 | 161.2 |
| BattleZone | 2360.0 | 37187.5 | 4088.0 | 13999.0 | 14416.0 | 11306.0 | 4700.0 | 2099.0 |
| Boxing | 0.1 | 12.1 | 11.56 | 16.43 | 21.2 | 18.75 | 44.14 | 44.5 |
| Breakout | 1.7 | 30.5 | 13.16 | 12.64 | 13.71 | 10.56 | 18.46 | 21.0 |
| ChopperCommand | 811.0 | 7387.8 | 778.0 | 977.7 | 879.2 | 1150.6 | 777.5 | 891.2 |
| CrazyClimber | 10780.5 | 35829.4 | 13182.2 | 17927.7 | 15852.8 | 13957.6 | 19082.8 | 31223.8 |
| DemonAttack | 152.1 | 1971.0 | 701.38 | 735.54 | 556.33 | 531.05 | 1074.29 | 2117.8 |
| Freeway | 0.0 | 29.6 | 22.32 | 27.79 | 30.49 | 24.54 | 26.92 | 27.1 |
| Frostbite | 65.2 | 4334.7 | 1856.64 | 2052.95 | 1409.34 | 763.64 | 1257.33 | 1802.3 |
| Gopher | 257.6 | 2412.5 | 374.28 | 476.02 | 433.92 | 445.88 | 711.14 | 839.4 |
| Hero | 1027.0 | 30826.4 | 5096.67 | 6496.56 | 5645.98 | 6332.89 | 6987.43 | 7007.2 |
| Jamesbond | 29.0 | 302.8 | 348.8 | 333.3 | 287.2 | 239.7 | 369.2 | 461.1 |
| Kangaroo | 52.0 | 3035.0 | 3773.2 | 3453.8 | 3589.0 | 4919.8 | 2830.8 | 1636.1 |
| Krull | 1598.0 | 2665.5 | 3612.06 | 3385.05 | 3483.26 | 3710.24 | 4379.46 | 5019.5 |
| KungFuMaster | 258.5 | 22736.3 | 20412.4 | 16380.3 | 8646.2 | 15188.4 | 12737.4 | 16105.0 |
| MsPacman | 307.3 | 6951.6 | 1210.02 | 1471.33 | 1226.5 | 1191.2 | 1362.05 | 1245.6 |
| Pong | −20.7 | 14.6 | −8.82 | −6.42 | −12.85 | 1.96 | −10.0 | −17.7 |
| Qbert | 11.5 | 7845.0 | 3776.3 | 3495.45 | 2648.4 | 2429.2 | 3492.18 | 3986.3 |
| RoadRunner | 68.4 | 42054.7 | 15696.6 | 11919.5 | 13045.6 | 13703.4 | 12939.5 | 15073.8 |
| Seaquest | 24.9 | 69571.3 | 525.88 | 519.2 | 394.24 | 383.8 | 645.26 | 635.9 |
| PrivateEye | 163.9 | 13455.0 | 100.0 | 96.04 | 100.0 | 99.4 | 100.13 | 100.0 |
| UpNDown | 533.4 | 11693.2 | 3690.74 | 4963.46 | 5596.2 | 4565.04 | 15463.98 | 66473.0 |
| IQM | 0.0 | 1.0 | 0.258 | 0.325 | 0.259 | 0.256 | 0.343 | **0.421** |
| Median | 0.0 | 1.0 | 0.277 | 0.327 | 0.247 | 0.193 | 0.291 | **0.347** |
| Mean | 0.0 | 1.0 | 0.476 | 0.501 | 0.463 | 0.498 | 0.660 | **0.933** |
| OG | 1.0 | 0.0 | 0.633 | 0.589 | 0.627 | 0.628 | 0.579 | **0.535** |

Table 8: **Mean trajectory scores on State-of-the-art methods.** We report the individual scores on the 26 Atari games. For IRIS, DreamerV3, EfficientZero, PlayVirtual, MLR, SR-SPR, and PLASTIC the results are averaged over 5, 5, 32, 15, 3, 10, and 5 seeds respectively.

| Game | EfficientZero | IRIS | DreamerV3 | PlayVirtual | MLR | SR-SPR:4 | SR-SPR:8 | PLASTIC[†]:8 | PLASTIC:8 |
|---|---|---|---|---|---|---|---|---|---|
| Alien | 808.5 | 420.0 | 1095.8 | 947.8 | 990.1 | 964.4 | 1015.5 | 1021.7 | 1368.6 |
| Amidar | 148.6 | 143.0 | 142.9 | 165.3 | 227.7 | 211.8 | 203.1 | 186.2 | 220.2 |
| Assault | 1263.1 | 1524.4 | 638.0 | 702.3 | 643.7 | 987.3 | 1069.5 | 821.5 | 1074.2 |
| Asterix | 25557.8 | 853.6 | 982.8 | 933.3 | 883.7 | 894.2 | 916.5 | 1184.7 | 1188.0 |
| BankHeist | 351.0 | 53.1 | 617.8 | 245.9 | 180.3 | 460.0 | 472.3 | 370.1 | 313.8 |
| BattleZone | 13871.2 | 13074.0 | 12800.0 | 13260.0 | 16080.0 | 17800.6 | 19398.4 | 15728.0 | 16868.0 |
| Boxing | 52.7 | 70.1 | 67.8 | 38.3 | 26.4 | 42.0 | 46.7 | 45.9 | 41.8 |
| Breakout | 414.1 | 83.7 | 18.9 | 20.6 | 16.8 | 26.1 | 28.8 | 21.9 | 20.9 |
| ChopperCommand | 1117.3 | 1565.0 | 400.0 | 922.4 | 910.7 | 1933.7 | 2201.0 | 799.4 | 871.2 |
| CrazyClimber | 83940.2 | 59324.2 | 71620.0 | 23176.2 | 24633.3 | 38341.7 | 43122.3 | 31652.6 | 47893.0 |
| DemonAttack | 13003.9 | 2034.4 | 545.0 | 1131.7 | 854.6 | 3016.2 | 2898.1 | 1562.9 | 2460.9 |
| Freeway | 21.8 | 31.1 | 0.0 | 16.1 | 30.2 | 24.5 | 24.9 | 29.4 | 30.4 |
| Frostbite | 296.3 | 259.1 | 1108.4 | 1984.7 | 2381.1 | 1809.9 | 1752.8 | 2152.5 | 2195.4 |
| Gopher | 3260.3 | 2236.1 | 5828.6 | 684.3 | 822.3 | 717.5 | 711.2 | 582.7 | 612.6 |
| Hero | 9315.9 | 7037.4 | 10964.6 | 8597.5 | 7919.3 | 7195.7 | 7679.6 | 11195.5 | 9000.7 |
| Jamesbond | 517.0 | 462.7 | 510.0 | 394.7 | 423.2 | 408.8 | 392.8 | 401.3 | 428.8 |
| Kangaroo | 724.1 | 838.2 | 3550.0 | 2384.7 | 8516.0 | 2024.1 | 3254.9 | 6218.2 | 2249.2 |
| Krull | 5663.3 | 6616.4 | 8012.0 | 3880.7 | 3923.1 | 5364.3 | 5824.8 | 5201.9 | 5647.6 |
| KungFuMaster | 30944.8 | 21759.8 | 29420.0 | 14259.0 | 10652.0 | 17656.5 | 17095.6 | 20839.6 | 19546.2 |
| MsPacman | 1281.2 | 999.1 | 1388.5 | 1335.4 | 1481.3 | 1544.7 | 1522.6 | 1662.0 | 1292.4 |
| Pong | 20.1 | 14.6 | 18.5 | −3.0 | 4.9 | −5.5 | −3.0 | 0.3 | −3.5 |
| Qbert | 13781.9 | 745.7 | 3117.9 | 3620.1 | 3410.4 | 3699.8 | 3850.6 | 4372.8 | 4967.2 |
| RoadRunner | 17751.3 | 9614.6 | 14036.0 | 13429.4 | 12049.7 | 14287.3 | 13623.5 | 16254.0 | 20709.0 |
| Seaquest | 1100.2 | 661.3 | 582.0 | 532.9 | 628.3 | 766.6 | 800.5 | 574.7 | 859.1 |
| PrivateEye | 96.7 | 100.0 | 1124.0 | 93.9 | 100.0 | 95.8 | 95.8 | 100.0 | 100.0 |
| UpNDown | 17264.2 | 3546.2 | 9234.0 | 10225.2 | 6675.7 | 91435.2 | 95501.1 | 34342.4 | 33203.3 |
| IQM | n/a | 0.501 | 0.497 | 0.374 | 0.432 | 0.544 | 0.589 | 0.583 | 0.571 |
| Median | 0.227 | 0.289 | 0.466 | n/a | n/a | 0.523 | 0.560 | 0.542 | 0.494 |
| Mean | 0.562 | 1.046 | 1.097 | n/a | n/a | 1.111 | 1.188 | 0.939 | 0.968 |
| OG | n/a | 0.512 | 0.505 | 0.558 | 0.522 | 0.470 | 0.452 | 0.448 | 0.461 |

Table 9: **Mean trajectory scores for each value of replay ratio (2, 4, 8).** We report the individual scores on the 26 Atari games. Here, the results are averaged over 5 random seeds.

| Game | PLASTIC[†]:2 | PLASTIC:2 | PLASTIC[†]:4 | PLASTIC:4 | PLASTIC[†]:8 | PLASTIC:8 |
|---|---|---|---|---|---|---|
| Alien | 1063.1 | 1032.0 | 1251.2 | 1138.3 | 1021.7 | 1368.6 |
| Amidar | 195.2 | 201.6 | 151.4 | 206.1 | 186.2 | 220.2 |
| Assault | 763.6 | 888.5 | 767.2 | 939.7 | 821.5 | 1074.2 |
| Asterix | 1007.6 | 1066.0 | 1098.1 | 1058.0 | 1184.7 | 1188.0 |
| BankHeist | 217.7 | 161.2 | 328.1 | 268.0 | 370.1 | 313.8 |
| BattleZone | 17767.0 | 2099.0 | 17762.0 | 9292.0 | 15728.0 | 16868.0 |
| Boxing | 39.1 | 44.5 | 53.6 | 46.4 | 45.9 | 41.8 |
| Breakout | 16.4 | 21.0 | 22.4 | 27.9 | 21.9 | 20.9 |
| ChopperCommand | 972.4 | 891.2 | 798.0 | 875.4 | 799.4 | 871.2 |
| CrazyClimber | 21967.5 | 31223.8 | 24615.0 | 42557.6 | 31652.6 | 47893.0 |
| DemonAttack | 1016.2 | 2117.8 | 1630.6 | 2402.2 | 1562.9 | 2460.9 |
| Freeway | 30.1 | 27.1 | 29.6 | 29.6 | 29.4 | 30.4 |
| Frostbite | 1582.0 | 1802.3 | 2119.4 | 1899.3 | 2152.5 | 2195.4 |
| Gopher | 616.4 | 839.4 | 635.2 | 811.6 | 582.7 | 612.6 |
| Hero | 6582.8 | 7007.2 | 10350.0 | 8118.2 | 11195.5 | 9000.7 |
| Jamesbond | 397.5 | 461.1 | 383.7 | 438.7 | 401.3 | 428.8 |
| Kangaroo | 2836.3 | 1636.1 | 6860.8 | 2386.6 | 6218.2 | 2249.2 |
| Krull | 4485.6 | 5019.5 | 4719.8 | 5266.4 | 5201.9 | 5647.6 |
| KungFuMaster | 11873.9 | 16105.0 | 17613.4 | 19828.6 | 20839.6 | 19546.2 |
| MsPacman | 1399.6 | 1245.6 | 1401.9 | 1457.2 | 1662.0 | 1292.4 |
| Pong | −6.3 | −17.7 | 0.1 | −7.1 | 0.3 | −3.5 |
| Qbert | 4006.1 | 3986.3 | 3794.1 | 4736.4 | 4372.8 | 4967.2 |
| RoadRunner | 15883.0 | 15073.8 | 17704.2 | 19487.8 | 16254.0 | 20709.0 |
| Seaquest | 576.1 | 635.9 | 692.8 | 537.0 | 574.7 | 859.1 |
| PrivateEye | 46.6 | 100.0 | 94.1 | 100.0 | 100.0 | 100.0 |
| UpNDown | 15470.0 | 66473.0 | 11398.0 | 51764.5 | 34342.4 | 33203.3 |
| IQM | 0.396 | 0.421 | 0.518 | 0.545 | 0.583 | 0.571 |
| Median | 0.425 | 0.347 | 0.517 | 0.407 | 0.542 | 0.494 |
| Mean | 0.702 | 0.933 | 0.858 | 1.002 | 0.968 | 0.939 |
| OG | 0.541 | 0.535 | 0.478 | 0.475 | 0.448 | 0.461 |

Table 10: **Mean trajectory scores on DrQ algorithm.** We report the mean trajectory scores on the DMC medium environments, evaluated on top of DrQ. For each random seed, the results are averaged over 10 trajectories at the end of training. The results are then averaged over 10 random seeds.

| Game | Base | SAM | LayerNorm | CReLU | Reset | PLASTIC |
|---|---|---|---|---|---|---|
| acrobot-swingup | 16.34 | 15.36 | 23.45 | 40.79 | 57.83 | 68.24 |
| cartpole-swingup_sparse | 79.03 | 197.91 | 164.57 | 321.83 | 740.0 | 691.16 |
| cheetah-run | 727.44 | 736.61 | 591.47 | 835.33 | 665.67 | 708.04 |
| finger-turn_easy | 207.02 | 129.92 | 325.21 | 221.0 | 232.76 | 379.69 |
| finger-turn_hard | 80.95 | 60.07 | 485.70 | 117.5 | 89.11 | 546.56 |
| hopper-hop | 284.78 | 255.84 | 277.48 | 293.52 | 284.69 | 254.62 |
| quadruped-run | 155.06 | 261.03 | 266.84 | 78.89 | 466.93 | 426.57 |
| quadruped-walk | 88.08 | 130.85 | 173.13 | 80.02 | 632.33 | 528.12 |
| reacher-easy | 505.08 | 738.80 | 926.94 | 914.3 | 957.96 | 879.71 |
| reacher-hard | 450.96 | 495.32 | 746.55 | 795.77 | 794.53 | 798.10 |
| walker-run | 632.93 | 639.94 | 652.41 | 643.24 | 564.95 | 632.55 |
| IQM | 213 | 278 | 412 | 338 | 514 | **565** |
| Median | 288 | 341 | 415 | 398 | 491 | **540** |
| Mean | 293 | 332 | 421 | 394 | 498 | **537** |

## D  Broader Impact

Our research possesses broader impacts in two different aspects. The first impact lies in the sample efficiency of Reinforcement Learning (RL) algorithms. We incorporate principles of preventing the loss of plasticity, leading to more efficient and adaptive solutions. This advance can improve RL's sample efficiency, opening new avenues for future research in this domain.

The second impact is on inclusivity and accessibility in RL. By decreasing data and computational demands, this work can aid underprivileged communities with fewer resources to participate in RL research and benefit from its many applications. This approach encourages a diverse range of perspectives and experiences in the field, enriching the community.

However, while our research has these positive implications, it is important to also acknowledge potential risks associated with RL technologies, particularly in robotics. We must continue to uphold ethical standards and prioritize safety to prevent misuse or harm that could arise from these advancements.

