# OpenReview forum: "PLASTIC: Improving Input and Label Plasticity for Sample Efficient Reinforcement Learning"
_NeurIPS.cc/2023/Conference — NeurIPS 2023 poster_

### Official Review · Reviewer_mzhA · 2023-07-04

**Soundness:** 4 excellent
**Presentation:** 4 excellent
**Contribution:** 2 fair
**Rating:** 7
**Confidence:** 4

**Summary:**


The paper studies the generalization and plasticity of deep reinforcement learning (RL) agents. The paper's primary contribution is a combination of the existing Sharpness-Aware Minimization (SAM) and Resetting mechanisms. The main insight is that combining SAM+Resets provides additive benefits: the former induces feature sparsity (and hence could be argued to improve generalization), while the latter yields more active units for the last layers (and hence could be argued to increase network plasticity). Moreover, the authors conduct an analysis on synthetic data and arrive at the conclusion that SAM improves adaptability to input changes, while resets mostly improve the ability to adapt to new targets. When SAM+Resets are applied to Data-Efficient Rainbow (DER) and Data-regularized Q (DrQ) on Atari 100k, the combination gives consistent improvements.


**Strengths:**

The main strength of the paper is a thorough empirical investigation:
- The majority of claims are supplemented with supporting evidence: for example, the authors report sharpness metrics (such as the trace of the Fisher matrix and the largest Hessian eigenvalue) to demonstrate that SAM increases the flatness of the solution (the result is non-trivial since, unlike in supervised learning, the loss surface is always changing in RL).
- The paper presents ablations for the design choices when applying SAM to RL algorithms (for example, whether to apply SAM for the whole network or for a subset of layers in DER or whether to apply SAM for both actor and critic networks in DrQ or individually).
- The SAM+Resets combination is thoroughly compared to other methods that were proposed to address plasticity loss (such as layer normalization and concatenated ReLUs).


**Weaknesses:**

There are several weak spots that prevent the reviewer from assigning a higher score:
- The SAM+resets combination is applied for the DER algorithm in Atari 100k, which is a weaker baseline than the SPR algorithm used in the original resets paper; the choice makes the reviewer wonder if the addition from SAM would be smaller for SPR and other more advanced algorithms.
- Figure 4 contains an encouraging demonstration of improvements in replay ratio scalability for DrQ on Atari-100k from adding SAM on top of resets. However, the paper does not have scalability results for the other settings (DrQ on DMC / DER on Atari 100k / SAC on DMC from proprietary states) making the reviewer question the generality of the insight.

There are also several minor weaknesses:
- Table 1 contains placeholder performance values 0.xxx for DER + CReLU
- Putting in bold the results for SAM + Reset in Table 2 might mislead a limited-attention reader. The difference between SAM + Reset 516 (441, 590) and Reset 514 (434, 590) is insignificant and should not be highlighted.

Addressing the outlined limitations might increase the score.


**Questions:**

The paper doesn’t study continuous control from proprietary states — what was the motivation for excluding the setting?

The combination of preliminaries in Section 3 and the experimental details in Section 5.1 is confusing: first, the authors give background about the Rainbow and Soft Actor-Critic algorithms, while later use their modifications, DER and DrQ, for actual experimentation. The reviewer suggests describing directly DER and DrQ.


**Limitations:**

While the paper presents evidence about the effects of SAM+Resets on network adaptability, an explanation is lacking: why SAM and Resets enhance input and label adaptability and, for example, not visa versa?
Likewise, the reviewer finds intriguing the result in Figure 2a (bottom left): on their own, both resets and SAM decrease the fraction of active units, while their combination yields a strong increase. Why?
Diving deeper into explaining the phenomena would increase the significance of the paper.

---

> ### Author Rebuttal · Authors · 2023-08-09
>
> Dear reviewer mzhA,
>
> Thank you for your constructive feedback. To address your concerns,
> - We've experimented with SAM on advanced algorithms.
> - We provide an explanation of the generality of our insight.
> - Rectified .xxx in Table 1.
> - The presentation in Table 2 will be revised to avoid misinterpretations.
> - Explained the rationale behind excluding state-based environments.
> - We plan to improve clarity between DER and DrQ.
>
> Please let us know if you have any further comments or feedback. We will do our best to address them.
>
>
> > **Question 4.1:**
> The reviewer wonders if the addition from SAM would be smaller for SPR and other more advanced algorithms.
>
> To clarify, our method primarily builds upon the DrQ, as detailed in Supplementary Section 5. Contrasting with SR-SPR, which couples DrQ with SSL and Reset, our approach couples DrQ with SAM and Reset. We have tried SR-SPR + SAM and found that the performance gains were indeed marginal. This can likely be attributed to the inherent generalization enhancements from the SSL objectives already present in SPR.
>
> Additionally, our SAM + Reset combination has been rigorously tested with advanced algorithms like BBF and SimTPR on the Atari-100k benchmark. The results, showcased in Table A.2 (attached pdf), confirm the robustness and efficacy of this synergy.
>
> In our revised manuscript, we'll accentuate these findings, emphasizing the distinctions and comparative benefits of our approach.
>
> > **Question 4.2:**
> The paper does not have scalability results for the other settings (DER on Atari 100k / DrQ, SAC on DMC) making the reviewer question the generality of the insight.
>
> For DER on Atari-100k: DER utilizes shallow and wide architecture and focuses on amplifying generalizable representations. Consequently, testing scalability on DER might detract from its foundational goal of enhancing the generalization of larger networks.
>
> For DrQ and SAC on DMC: Initially, we surmised that DMC's limited generalization advantages stemmed from lower input non-stationarity. Yet, our DMC-GB experiments on Table A.3 (attached pdf) and subsequent comparisons between Atari and DMC revealed that DMC indeed possesses a greater input non-stationarity.
>
> In addition, DMC employs rigorous reset protocols: DrQ resets almost 50% of its network, and SAC resets it entirely, as detailed in [1]. These comprehensive resets inherently counter input non-stationarities, consistently restoring the system to an initial state. As [1] has demonstrated the efficacy and scalability of such resets, we believe this further validates the relevance and breadth of our insights.
>
> Our further tests included scaling SAM+Reset on SimTPR, a 30-layer CNN model. As presented in Table A.1, the synergy of SAM and Reset is evident at a replay ratio of 2 and retains its potency when increased to 4. Recent findings from BBF also highlight the harmonious interplay between generalization (via strong L2 regularization and Self-Supervised objectives) and plasticity (via resets), across replay ratios from 2 to 8.
> In light of these, we remain confident in the broader applicability of our insights.
>
> [1] Sample-Efficient RL by Breaking the Replay Ratio Barrier, D’Oro et al., ICLR 2023.
>
> > **Question 4.3:**
> Table 1 contains placeholder performance values 0.xxx for DER + CReLU
>
> Thank you for pointing out the placeholders. The updated values for DER + CReLU and DER + LayerNorm, previously marked as “0.xxx,” are provided in Supplementary Section 6, highlighted in red. We apologize for any inconvenience this caused.
>
> > **Question 4.4:**
> Putting in bold the results for SAM + Reset in Table 2 might mislead a limited-attention reader.
>
> We agree that emphasizing the results for SAM + Reset could mislead readers given the minimal difference with just the Reset. We will present the results without bolding.
>
> > **Question 4.5:**
> The paper doesn’t study continuous control from proprietary states — what was the motivation for excluding the setting?
>
> We emphasized image-based environments due to their inherent non-stationarity and the pronounced distribution shifts. In these tasks, even minor agent actions can drastically alter pixel distributions, demanding robust generalization. Conversely, state-based scenarios generally present more consistent input distributions, making them less challenging from a generalization perspective.
>
> We'll elucidate this choice further in our revised manuscript for clarity.
>
> > **Question 4.6:**
> The combination of preliminaries in Section 3 and the experimental details in Section 5.1 is confusing. The reviewer suggests describing directly DER and DrQ.
>
> Thank you for pointing out the potential confusion. DER and DrQ, while rooted in similar foundational algorithms, differ in aspects like model architecture and data augmentation. In our revised manuscript, we'll refine the presentation, explicitly highlighting these differences to ensure a clear understanding.
>
> > **Question 4.7:**
> Why do SAM and Resets enhance input and label adaptability and not visa versa? In Figure 2a, both resets and SAM decrease the fraction of active units in the head, while their combination yields a strong increase. Why?
>
> On the distinction in adaptability enhancements: SAM, by its design, results in sparser features, which may not readily allow for flexibility in adapting to shifting label dynamics. Reset, on the other hand, appears to naturally assist in label adaptability by preventing overfitting to evolving label relationships. However, intuitively, its precise contribution to input adaptability is less evident.
>
> Regarding the combined effect observed in Figure 2a: SAM's sparsity might necessitate a more active head unit to decipher diverse feature combinations, and when paired with Resets, this effect amplifies, leading to an overall rise in active units for the head. However, we acknowledge that this is a hypothesis, and further investigations are needed to solidify this understanding.

---

> > ### Comment · Reviewer_mzhA · 2023-08-11
> >
> > Thank you for the clarifications. In response to the rebuttal, I have updated my score to 7.
> > I am eager to see follow-up works that even further deepen the understanding of the differences between input and target non-stationarities in RL.

---

> > > ### Author Response · Authors · 2023-08-15
> > >
> > > Thank you for your constructive feedback and for updating the score. We appreciate your insights and look forward to further exploring this topic in our future work.

---

### Official Review · Reviewer_dLhS · 2023-07-05

**Soundness:** 3 good
**Presentation:** 4 excellent
**Contribution:** 3 good
**Rating:** 6
**Confidence:** 4

**Summary:**

This paper presents a new method to enhance sample efficiency in reinforcement learning, by integrating two existing techniques: sharpness-aware minimization (SAM) and weight resetting. It shows that SAM and resetting work in a complementary way where SAM addresses input adaptability and resetting addresses label adaptability. Experiments on Atari100K and DM control demonstrate the effectiveness of the proposed method.

**Strengths:**

- This paper is clear, well-written, self-contained, and enjoyable to read.
- While the proposed method simply combines two existing approaches, such a combination is novel and well justified by the synthetic experiments and ablation studies.
- The experiments are comprehensive and well executed.

**Weaknesses:**

- The authors mention that the relatively small improvement in DMC might be due to the reduced visual variety in DMC. The [DMControl Generalization Benchmark](https://github.com/nicklashansen/dmcontrol-generalization-benchmark) augments DMC with rich visual variety, which could be a good testbed to validate the claim.

**Questions:**

Apart from the points raised above, I have some questions and comments, and would like to hear the authors' feedback:
- Table 1: Some entries missing ("xxx" in the table).
- Line 128: The sentence ("..., which encourages to Here, ...") is confusing.
- I notice the authors have cited [10] in the related works. I am wondering if the authors have tried the soft resetting strategy on Atari, since it has been shown to be effective.
- It would be better to add error bars in Figure 1(right), Figure 4, and Figure 5.

**Limitations:**

The limitations of the proposed method have been adequately discussed in Section 6.

---

> ### Author Rebuttal · Authors · 2023-08-09
>
> Dear reviewer dLhs,
>
>
> Thank you for your constructive feedback. To address your concerns,
> - We explored the DMControl Generalization Benchmark (DMC-GB). Our findings from this extended study suggest that DMC exhibits a large degree of input non-stationarity than we initially presumed.
> - We rectified discrepancies in Table 1, improved clarity in various sections, and enhanced our figures.
> - In a broader perspective, we've also conducted rigorous tests in synthetic environments and experimented with SAM + Reset on more advanced algorithms. Further detail is described in our general response.
>
> Please let us know if you have any further comments or feedback. We will do our best to address them.
>
>
> > **Question 3.1:**
> The authors mention that the relatively small improvement in DMC might be due to the reduced visual variety in DMC. The DMControl Generalization Benchmark augments DMC with rich visual variety, which could be a good testbed to validate the claim.
>
> Thank you for bringing up the DMControl Generalization Benchmark (DMC-GB). In contrast to the traditional approach of DMC-GB—training on clear inputs and testing on noisy ones—we opted to train and test on noisy inputs. This approach aligns with our intention of illustrating the strength of different generalization techniques against input non-stationarity.
>
> Our experimental results in Table A.3 (attached pdf) indicate that when paired with a reset, SAM performs comparably to Adam. Without the reset, however, SAM surpasses Adam. An intriguing observation was that the DMC environment displayed a higher degree of input non-stationarity compared to Atari, even without the background noise introduced in DMC-GB. This challenges our initial hypothesis, suggesting that handling severe input non-stationarity might be better suited for DMC.
>
> Then, if DMC inherently demands addressing input non-stationarity, why does the reset strategy perform so well? The context lies in the different reset strategies between environments. In the original reset paper referenced as [1], DMC employs an aggressive reset strategy, reinitializing nearly 90% of its network for DrQ and 100% of its network on SAC. Instead, in environments like Atari, 50% of the network is reinitialized. This aggressive approach, highlighted in [1] effectively manages input non-stationarity, potentially overshadowing the benefits of SAM when paired.
>
> Our research primarily focuses on resetting the final few layers. However, the efficacy of broad resets in tackling input non-stationarity is evident in our synthetic experiments, where Reset (B, H) showcases effectiveness in Figure A.1 (attached pdf) While these aggressive resets worked well for DMC, we believe this strategy is not always optimal as evidenced by the results of Reset (B, H) from Atari in Table A.1. Extensive reinitialization forces the model to relearn, and as networks grow larger, this becomes more challenging.
>
> Thus, we believe blending different strategies to combat input non-stationarity is essential to further enhance the performance of both DMC and DMC-GB.
>
> [1] The Primacy Bias in Deep Reinforcement Learning., Nikishin et al., ICML 2022.
>
> > **Question 3.2:**
> Table 1: Some entries missing ("xxx" in the table).
>
> The updated values for DER + CReLU and DER + LayerNorm, previously marked as “0.xxx,” are provided in Supplementary Section 6, highlighted in red. We apologize for any inconvenience this caused.
>
>
> > **Question 3.3:**
> Line 128: The sentence ("..., which encourages to Here, ...") is confusing.
>
> Thank you for pointing this out. We found out that the sentence was inadvertently truncated. The correct sentence should read:
>
> "... which encourages a model parameter $w$ to find a smoother region of the loss landscape.”
>
> We will ensure it is corrected in the revised manuscript.
>
>
> > **Question 3.4:**
> I noticed the authors have cited [10] in the related works. I am wondering if the authors have tried the soft resetting strategy on Atari since it has been shown to be effective.
>
> Thank you for highlighting the soft resetting strategy referenced in [10].
>
> Indeed, we have experimented with the soft reset approach on the Atari benchmarks, as detailed in our appendix section 5. Our findings indicate that when the replay ratio is set at 2 (the standard setup in our primary experiments), the use of the soft reset (i.e., Shrink & Perturb) tends to degrade performance. However, as we increment the replay ratio, the advantages of the soft reset became notably evident, even surpassing the results of well-known state-of-the-art algorithms.
>
>
> > **Question 3.5:**
> It would be better to add error bars in Figure 1(right), Figure 4, and Figure 5.
>
> We will incorporate error bars in Figures 1(right), 4, and 5 in the revised manuscript.

---

> > ### Comment · Reviewer_dLhS · 2023-08-12
> > **Feedback**
> >
> > Thank the authors for the detailed reply and additional experiments. I have one comment regarding Question 3.1 and expect the authors can take it into consideration when preparing the next version. The results show that "SAM + Resets" performs similarly to or worse than "SAM" on DMC and DMC-GB. The authors now give a new explanation, attributing it to the "aggressive reset strategy". If the benefits of SAM are potentially overshadowed by resetting on DMC, then the authors should make it clear in the paper (especially abstract) and modify the claims like "Extensive empirical studies on the Atari-100k and DeepMind Control Suite benchmarks demonstrate that this combined usage yields sparse, generalizable features and a dense, plastic policy.".

---

> > > ### Author Response · Authors · 2023-08-15
> > >
> > > We appreciate your feedback regarding Question 3.1. We agree with your observation and the importance of clarity around the impact of the "aggressive reset strategy". We will ensure that this clarification is adequately reflected in the revised manuscript.

---

### Official Review · Reviewer_GE2W · 2023-07-06

**Soundness:** 2 fair
**Presentation:** 2 fair
**Contribution:** 3 good
**Rating:** 7
**Confidence:** 5

**Summary:**

This work studies the role of generalization and plasticity in sample-efficient deep RL. This paper proposes sharpness-aware minimization (SAM) to improve generalization in RL. And it provides details on how to use SAM with deep RL algorithms like SAC and Rainbow. Empirical evaluation shows that combined usage of SAM and periodically resetting the last few layers of the network improves sample efficiency in the Atari-100k benchmark. On the other hand, adding SAM does not provide any benefit over resetting in Deepmind Control Suite.

**Strengths:**

This work tackles the critical problem of sample efficient reinforcement learning. Sample efficiency is particularly important for applications where exploration is risky and expensive, such as healthcare.

The paper proposes using SAM to improve the generalization of deep RL algorithms. SAM has not been used in RL before, so the details regarding the usage of SAM in RL are valuable.

The paper attempted to study the role of generalization and plasticity in a simple continual supervised learning problem. This problem does not have the confounders found in RL, so it can improve our understanding of the underlying phenomena.

The results on the Atari-100k benchmark show that combining SAM with resets improves the performance over just using SAM or resets. This exciting result shows that SAM can improve the sample efficiency of deep RL.

**Weaknesses:**

This paper contains interesting ideas and promising experimental results. The paper's central claim is that generalizability and plasticity constitute different roles in improving performance (lines 6-7). However, the experiments performed in the paper do not adequately test this claim. There are significant issues with the experiments in Sections 4 and 5.2.

The paper claims that "we find that incorporating both generalization and plasticity is crucial for improving the model's ability to adapt to new data (Section 4)" (lines 56-57). Unfortunately, the experiments in Section 4 have the following major problems:

1. The paper claims that generalization and plasticity constitute different roles. To represent generalization, it used SAM, and to represent plasticity, it periodically reset the last few layers. Then two experiments were performed, one where the input changes over time and the other where the labels change. The results show that resetting the last few layers does not help when the input changes. But that is not surprising. Resetting the last few layers only injects plasticity into the last few layers. However, the conclusions drawn from this result are too general. The paper says that plasticity does not help when the input distribution changes, but we can only draw this conclusion if the paper contains results for methods that inject plasticity into the whole network.

The experiments in Section 4 should include methods that inject plasticity into the full network. This means including methods that reset the last few layers and use CReLUs in the full network, or reset the last few layers and use LayerNorm after all the layers, or using selective reinitialization methods like Continual Backprop or ReDo. From the current results, it is unclear if SAM provides any benefits over plasticity injecting methods.

2. The experiments were only performed for five random seeds (line 73 of the appendix), which raises questions about the statistical significance of these results. These experiments are performed on CIFAR, so they are probably not computationally expensive. I think it should be possible to perform more runs, say 30, to increase the confidence in these results. The paper also needs to report how the confidence interval is calculated in Figure 1.

3. The hyperparameters are not properly tuned for the input adaptation experiment. The experiment used a learning rate of 0.001, which was the smallest one that was tested. Even lower learning rates need to be tested.

Similar problems exist with the experiments in Section 5.2.
1. Again, the resetting methods used in these experiments only inject plasticity in the last few layers. The effect of injecting plasticity in the full network needs to be tested before including SAM.

2. The paper proposes L2 regularization as a baseline for SAM. However, it is not tested in the experiments. Experiments with DER show that L2 performs better than SAM, so it is puzzling why L2+Reset is not compared to SAM+Reset. L2+plasticity-preserving-methods should be compared with SAM+Reset.

Besides these major concerns, the clarity of the writing can be improved. For example, the sentence on line 128 needs to be completed. Spending more space to explain section 3.2 will also be valuable for the community.

**Questions:**

1. The paper uses two terms, "adaptability" and "plasticity." To me, these words have very similar meanings. Plasticity means the ability to learn from new data, and adaptability means being able to fit new data. Can you please explain in what sense are you using these words and what is the difference between the two? If they have a similar meaning, using just one word in the paper might be better. Currently, the use of these two words causes confusion in section 4. Maybe in the conclusion of Section 4, it is better to say that SAM improves plasticity when there is input non-stationarity and resetting the last few layers improves plasticity when the labels change.

2. What is the confidence interval reported in Figure 2? Is it the 95% bootstrapped confidence interval?

**Limitations:**

The key limitation of the paper is that the experiments did not test the central claims of the paper. To claim that both plasticity and generality are important for improving adaptability, methods that inject plasticity into the whole network should be tested.
Including the following baselines in Section 4 with proper hyper-parameter turning and more random seeds will significantly improve the completeness of the claims.
1. CReLU
2. LN
3. L2 (properly tuned) + Reset
4. CReLU + Reset
5. LN + Reset

Similarly, the following baselines in Section 5.2 need to be added.
1. CReLU + Reset
2. LN + Reset
3. L2 + Reset
4. L2 + best combination of CReLU, LN, Reset
5. SAM + best combination of CReLU, LN, Reset


The paper contains interesting ideas and promising results. However, the current empirical evaluation is incomplete and does not support the central claim of the paper. Unfortunately, I can not recommend accepting the paper in its current form. But, I'm willing to change my score if the authors can include all the relevant baselines for experiments in Sections 4 and 5.2 and show that the conclusions still hold.

EDIT: I have update my score based on the new experiments and changes in the main message of the paper.

---

> ### Author Rebuttal · Authors · 2023-08-09
>
> Dear reviewer GE2W,
>
> We appreciate your constructive guidance. Based on your feedback,
> - We've refined our Synthetic experiments.
> - We explored the synergies of input and label adaptation techniques in RL experiments.
> - Line 128 has been corrected for clarity.
> - We plan to provide a detailed explanation of SAM in Section 3.
> - We clarified the terminology focusing on “plasticity”.
>
> Please let us know if you have any further comments or feedback. We will do our best to address them.
>
> > **Question 2.1:**
> The experiments in Section 4 have the following major problems:
> - The hyperparameters are not properly tuned.
> - The experiments in Section 4 should include methods including  {CReLU, LN, L2 (properly tuned) + Reset, CReLU + Reset, LN + Reset}.
> - Only five random seeds were used.
> - Needs to report confidence intervals.
>
> Thank you for your observations. We've undertaken a rigorous revision of our synthetic experimental setup:
>
> - Tuning Learning Rate & Weight Decay: Recognizing the criticality of tuning for learning rate and weight decay in non-stationary datasets, we've executed an exhaustive search across {0.1, 0.01, 0.001, 0.0001, 0.00001} for both. We decided to incorporate weight decay as a base setup as we found it was important to reduce the variance of each individual run.
> - Broadening Baselines: We broadened our benchmarks to include {LN (B+H), LN (B), SAM (B+H), ReDO (B+H), CReLU (B+H), CReLU (H), Reset (H)}, with B and H denoting backbone and head network parts. For ReDO we tuned the dormant threshold from {0.2, 0.1, 0.05, 0.02, 0.01}. In addition, we further investigated the synergistic interactions of these baselines.
> - Increasing Random Seeds: We increased the number of random seeds from 5 to 30.
> - Reporting Confidence Interval: For each method, we report the 95% confidence interval.
>
> Figure A.1 (attached pdf), reveals a clear bifurcation of algorithms excelling in either input adaptation or label adaptation. For input adaptation, LN(B) and SAM(B+H) demonstrated prominent efficacy. Conversely, for label adaptation, Reset (H) and CReLU (H) were effective. Exploring the synergies of combined approaches offered interesting insights. The best performance across both scenarios was achieved either by strategically blending methods that exhibited proficiency in both input and label adaptation or by incorporating all of the methodologies.
>
> We hope these comprehensive experiments address your concerns and illustrate the robustness of our message.
>
>
> > **Question 2.2:**
> Similar problems exist with the experiments in Section 5.2.
>
> - The baselines used in Synthetic Experiments need to be added.
> - L2+plasticity-preserving-methods should be compared with SAM+Reset.
>
> We extended our experiments in the context of a sample-efficient RL setup, specifically with DrQ on the Atari-100k benchmark. From Table A.1(attached pdf), we observed a pronounced synergy when mixing input adaptation with label adaptation techniques. On the other hand, when we concentrated exclusively on either input or label adaptation, we only observed marginal enhancements.
>
> For L2 regularization, we conducted careful tuning both individually and in combination with other methods, but its performance was found to be suboptimal. Therefore, we prioritize other methods for exploration. Regarding the specific combinations like Reset(B+H) and LN(B+H) suggested by the reviewer: given the time constraints, we didn't delve into these. However, we are willing to explore these avenues in our revised manuscript.
>
> To conclude, our extended experiments reaffirm our initial hypotheses, and we're excited to further refine our work based on the insights provided.
>
>
> > **Question 2.3:**
> The sentence on line 128 needs to be completed.
>
> We found out that the sentence was inadvertently truncated. The complete sentence should be as follows.
>
> “... which encourages a model parameter $w$ to find a smoother region of the loss landscape.”
>
> We will ensure it is corrected in our revised manuscript.
>
>
> > **Question 2.4:**
> Spending more space to explain section 3.2 will also be valuable for the community.
>
> Thank you for your suggestion. We understand that Sharpness Aware Minimization (SAM) might be less familiar to the reinforcement learning community. We will expand on Section 3.2 in the revised manuscript to ensure clarity of the methodology.
>
>
> > **Question 2.5:**
> To me, "adaptability" and "plasticity" have very similar meanings. If they have a similar meaning, using just one word in the paper might be better.
>
> We concur with your observation that "plasticity" has been ambiguously defined in the reinforcement learning literature. However, upon reviewing the definitions from various literature from continual learning [1,2] and neuroscience [3,4], it does appear that using "plasticity" in the same context as "adaptability" would be more natural.
>
> To clarify the definitions:
> - We redefine "Plasticity" as the model's ability to adapt.
> - "Input Plasticity" refers to the model's capability to adjust to input non-stationarity or changes in $p(x)$.
> - "Label Plasticity" refers to the model's capability to adjust to label non-stationarity or shifts in $p(y|x)$.
> - We plan to retitle our paper as "Enhancing Input and Label Plasticity for Sample Efficient Reinforcement Learning."
>
> We believe this change in terminology and definition will provide more clarity and prevent confusion throughout the paper.
>
> [1] A study on the plasticity of neural networks. Berariu et al., arXiv 2021
>
> [2] Continual backprop: Stochastic gradient descent with persistent randomness. Dohare et al., arXiv 2022.
>
> [3] Neuroplasticity. Costandi, Moheb., MIt Press 2016.
>
> [4] Neuroplasticity: New biochemical mechanisms. Reznikov et al., Springer Healthcare, 2020.
>
>
> > **Question 2.6:**
> What is the confidence interval reported in Figure 2?
>
> It is the 95% bootstrapped confidence interval. We will clarify this in the revised manuscript.

---

> > ### Comment · Reviewer_GE2W · 2023-08-15
> >
> > Dear authors, thank you for your reply and for performing the additional experiments. The new experiments help with supporting the key claims of the paper.
> >
> > The new experiments and more runs for the experiments in Section 4 support the claim that SAM adds something that none of the existing solutions to the loss of plasticity provide. The new experiments also alleviate most of my concerns about the empirical rigour of the experiments.
> > I suggest authors also include an experiment in the final manuscript that shows what happens when individual components are removed from the "ALL" baseline in Figure A.1. In other words, it would be good to also have the performance of ALL - SAM(B+H), ALL-Reset(H), ALL-CReLU(H), ALL-LN(B) in figure A.1. Of course, these results can go in an appendix.
> >
> > The new experiments in Section 5.2 are more mixed. The difference between the performances of SAM(B, H) + Reset(H) and LN(B) + Reset (H) is not statistically significant. Similarly, there the difference between the performances of ALL (the last row of Table A.1) and ALL - SAM (B, H) is not statistically significant. So, it is unclear if SAM provides any benefit over existing plasticity-preserving methods. With that said, I agree with the authors that the results show that techniques that enhance input and output plasticity are complementary, and we can obtain the best results by combining methods that address both of these independently.
> >
> > As a side note, I think the authors should not bold methods in tables that are not statistically better than all other methods; I think it is a little misleading. There are probably better ways to show which methods perform the best, maybe using heatmaps.
> >
> > I thank the authors for checking the definition of plasticity in the continual learning and neuroscience literature. I like their plan to use terms like 'input plasticity' and 'output plasticity' and change the paper's title to reflect their main message more adequately. The empirical results support the new title, which was not the case for the previous title and main message.
> >
> > I want to point out that changing the terminology and main message (that input and output plasticity play complementary roles) requires significant rewriting. For example, parts of the abstract that currently discuss separate roles of generalization and plasticity have to be changed to discuss complementary roles of input and output plasticity. However, I feel confident that the authors will do a good job of adequately rewriting the paper to reflect the key message. Based on the new results that improve the empirical rigour of the experiments and the change of the main message (as reflected by the new title), I have updated my score to accept the paper.

---

> > > ### Author Response · Authors · 2023-08-16
> > >
> > > Thank you once again for your insightful feedback and rigorous examination of our manuscript.
> > >
> > > Through the extensive experiments based on your recommendations, our fundamental message on the intertwined roles of input and output plasticity has been crystallized. This clarity has deepened our understanding and underscored our primary message. Consequently, we are eager to revise our paper's narrative, accentuating the complementary nature of input and label plasticity.
> > >
> > > In parallel, these extensive experiments have provided a more nuanced perspective on SAM. While it certainly presents value, its impact may not be as profound as we initially stated. We recognize the nuance you've highlighted and commit to presenting SAM in a more toned-down manner.
> > >
> > > Your feedbacks were invaluable to our work, and we are dedicated to refining our manuscript.
> > >
> > > Warm regards, Authors of Submission 12455.

---

### Official Review · Reviewer_1mez · 2023-07-07

**Soundness:** 4 excellent
**Presentation:** 4 excellent
**Contribution:** 4 excellent
**Rating:** 8
**Confidence:** 3

**Summary:**

Sample efficiency in RL is desirable to reduce computational and data collection costs, and is particularly critical to data-limited domains.
While off-policy methods can improve sample efficiency by training multiple passes over the same data, it faces challenges due to overfitting, which makes it harder for the model to adapt to new data.
This paper argues that to address the problem, it is important to tackle both the generalization and plasticity of the model, and proposes a method that achieves this.
The method uses Sharpness-Aware Minimization (SAM) to improve the model’s generalization, and a reset mechanism that periodically reinitializes the final layers of the model to inject plasticity.
Using a synthetic supervised learning experiment, the authors show that SAM helps the model better adapt to new inputs, and the reset mechanism helps it to adapt better to new labels.
These improvements in adaptability enables off-policy methods to better utilize multiple updates on data, improving sample efficiency.

The authors evaluate their method on Atari 100k and DMC-M, applying them to the DER and DrQ learning algorithms that are designed for sample efficient learning.
The results show that both DER and DrQ gain significant performance improvements when equipped with SAM + reset, compared to other methods that aim to improve generalization or plasticity alone.
The authors further perform ablation studies.
Notably, they find that SAM + CRelu performs almost as well as SAM + reset, which suggest that it is the combination of improving generalization and adaptability that leads to the performance gains, rather than specific synergies between SAM and reset.
This highlights that the value of their contribution, which does not lie in merely combining two known techniques, but in discovering a synergistic dynamic between generalization and plasticity for sample efficient RL.

**Strengths:**

- The writing is clear, and figures are intuitive and well-presented. The authors' claims are modulated and supported by the experiments. The logic of the paper flows very well; many questions that I had while reading were quickly addressed in subsequent sections.
- The paper studies an important topic that has broad-ranging technical and environmental impacts. Sample efficient learning is not only an interesting technical problem, but also directly contributes to decreasing the environmental footprint of our field. In the age of increasingly large models trained with increasingly large amounts of data and compute, this is a critical issue.
- The solution that the authors propose is simple yet elegant, and can be widely applied as it requires little change to model architecture. Furthermore, beyond showing strong empirical results for their approach, the authors also provide a hypothesis on how the underlying principles of generalization and plasticity achieve a synergistic dynamic, and support it with synthetic experiments.

**Weaknesses:**

- Both SAM and the reset mechanism are known techniques, and applying them in conjunction to achieve good results is not by itself sufficient for novelty. However, in the ablation studies, the authors also demonstrate that SAM + CRelu performs almost as well as SAM + reset. This strengthens their claim that it is the synergistic relationship between generalization and plasticity, not the specific method to improve these properties, that leads to the performance gains, which is novel. Yet, just testing one alternative combination does not seem sufficient to support the claim.
- The experiments that study the impact of generalization and plasticity on the model's learning dynamics is synthetic and on supervised learning, which may not be an accurate approximation of the dynamics faced during RL. For example, uniform-randomly changing labels may not accurately simulate the moving targets problem faced in deep Q-learning.
- The base learning algorithms that the authors evaluate with, DER and DrQ, are not state-of-the-art for data-limited RL. For example, EfficientZero reports achieving a normalized median of 1.090 and normalized mean of 1.943 on Atari 100k, which is better than the results shown in this paper.
- Several fields appear to be missing in Table 1?

**Questions:**

- Have you tested other combinations of methods to improve generalization and plasticity? For me, the main contribution is not SAM + reset, but the insight that generalizability and plasticity play separate roles in improving adaptability, and combining methods that improve each of them can lead to further performance gains; SAM + reset is just one instantiation used to verify your hypothesis. Ideally, I'd like to see that the same pattern holds on other combinations beyond SAM + reset and SAM + CRelu.
- Why did you select DER and DrQ as your base learning algorithms? Have you experimented with applying your method to stronger-performing algorithms, and would the same performance gains you achieved on DER and DrQ also hold there?


**Limitations:**

The authors are forthcoming with the main limitations of this work, and explained them clearly.

---

> ### Author Rebuttal · Authors · 2023-08-09
>
> Dear reviewer 1mez,
>
> We appreciate your insightful questions and positive support. We have provided a detailed response to the comments which includes additional experiments on finding different synergetic combinations and integration of SAM + Reset on the advanced algorithms. Please let us know if you have any further comments or feedback. We will do our best to address them.
>
> > **Question 1.1:**
> Have you tested other combinations of methods to improve generalization and plasticity? For me, the main contribution is the insight that generalizability and plasticity play separate roles in enhancing adaptability.
>
> Thank you for highlighting our paper's key insight. To relieve the reviewer’s concern, we did investigate various method combinations. As seen in Figure A.1 and Table A.1 (attached pdf), LN(B) and SAM(B+H) consistently excel in input adaptation, while CReLU(H) and Reset(H) stand out for label adaptation.
>
> When combining these methods, we made some insightful observations in both synthetic and reinforcement learning experiments:
>
> - CReLU + Reset showed moderate improvements.
> - SAM + LN also yielded moderate improvements.
> - Yet, pairing any input adaptation method (either SAM or LN) with any label adaptation method (CReLU or Reset) consistently produced robust results.
> - A comprehensive combination of all these methods further enhanced performance.
> These findings reinforce our notion of the distinct yet complementary roles of generalizability and plasticity in enhancing adaptability.
>
> > **Question 1.2:**
> For synthetic experiments, uniform-randomly changing labels may not accurately simulate the moving targets problem faced in deep Q-learning.
>
> Due to the inherent complexities of learning dynamics in RL, we simplified our experiments using a supervised learning setup. We acknowledge your concerns regarding the alignment of our "Label Adaptation" scenario with RL's dynamics, especially in deep Q-learning.
>
> In Q-learning, the agent undergoes two primary label adaptation scenarios: 1) the continual change of the best action as new data is received (akin to changing labels), and 2) the alteration of target Q-values (resembling noisy labels). We recognize the importance of both scenarios but decided to focus on the former scenario to keep the synthetic experiment manageable.
>
> We concur that our current design might not entirely capture the dynamics of RL. Exploring a more realistic synthetic experiment is indeed an exciting direction for future research. We will point out such limitations in our manuscript.
>
>
> > **Question 1.3:**
> Have you experimented with applying your method to stronger-performing algorithms?
>
>
> We have explored the applicability of our method beyond just DER and DrQ. Specifically, our experiments encompass BBF, a state-of-the-art algorithm that learns from scratch on the Atari-100k benchmark, and SimTPR, which leverages pre-trained representations for Atari. In both instances, combining SAM and Reset consistently outperformed using either method in isolation. Detailed empirical results can be found in Table A.2 (attached pdf). We appreciate your insightful suggestion.
>
>
> > **Question 1.4:**
> Several fields appear to be missing in Table 1.
>
> Thank you for pointing out the placeholders. The updated values for DER + CReLU and DER + LayerNorm, previously marked as “0.xxx,” were provided in Supplementary Section 6, highlighted in red. We apologize for any inconvenience this caused.

---

> > ### Comment · Reviewer_1mez · 2023-08-15
> >
> > Thank you for addressing my concerns. Given the robustness of the results, its novelty and broad impact, I have increased my score to an 8. I believe this work deserves to reach a wide audience.

---

> > > ### Author Response · Authors · 2023-08-16
> > >
> > > We deeply appreciate your positive evaluation and recognition of our work's potential impact. Your insights, conveyed through the review, serve as both guidance and motivation for our subsequent endeavors in this research direction. Thank you.

---

### Author Rebuttal · Authors · 2023-08-09

We sincerely appreciate all four reviewers for their constructive and insightful comments.

The reviewers recognized the strengths of our paper as follows:
- A fresh insight into the dissection of generalization and plasticity (Reviewer 1mez, GE2W).
- The introduction of a synergistic solution: SAM + Reset, which seamlessly integrates without any architectural modifications (Reviewer dlhS, mzhA).

We recognize that reviewers have highlighted several key points for us to address:
- Ensuring the robustness of our synthetic experiments (Reviewer GE2W).
- Delving deeper into explored synergies (Reviewer 1mez, GE2W).
- Investigating the approach's applicability to advanced algorithms (Reviewer 1mez, mzhA).
- Extension to various domains (Reviewer dlhS).

Below, we address each of these points in the following sections. Please refer to the attached PDF for detailed figures and tables.

> **Refinements to the synthetic experiments**

In response to feedback, we have:

- Adjusted learning rates and weight decay from a wide range of values.
- Broadened our experimental baselines, including ReDO [1]. For clarity: "B" signifies the network's backbone and "H" indicates its head. Our findings spotlight the efficacy of LN(B) and SAM(B+H) for input adaptation, and the efficacy of CReLU(H) and Reset(H) for label adaptation. Moreover, the interplay between these methods has been thoroughly investigated.
-Increased random seeds from 5 to 30.

As illustrated in Figure A.1, we found that

- Combining SAM(B+H) with LN(B), and pairing CReLU(H) with Reset(H), resulted in incremental improvements.
- Merging input-focused methods like SAM(B+H) or LN(B) with label-focused counterparts, such as CReLU(H) or Reset(H), consistently delivered impressive outcomes.
- Integrating all the methods together yielded the most significant enhancements.

These results vividly underscore the synergy between generalization and plasticity, reinforcing our research's relevance.

[1] The Dormant Neuron Phenomenon in Deep Reinforcement Learning. Sokar et al., ICML 2023



> **Deeper Exploration of Synergies**

Responding to the feedback received, we've intensified our investigation into the potential combinations of techniques within both synthetic and RL settings, as described in Figure A.1 and Table A.1.

Our in-depth analysis revealed that while methods tailored for input and label adaptability operate distinctively, they produce a remarkable synergy when fused. We believe this comprehensive exploration solidifies the importance of our contributions.

> **Extension to Advanced Algorithms**

Moving beyond the realms of DER and DrQ, we have ventured into more advanced algorithms:

- BBF [2]: Recognized as a state-of-the-art algorithm for the Atari-100k.
- SimTPR [3]: An algorithm that leverages pretrained representations for Atari.

Table A.2 summarizes our findings:

- BBF: Utilizing SSL + Reset or SAM + Reset distinctly outperforms isolated implementations.
- SimTPR: The amalgamation of SAM and Reset magnifies results compared to their standalone use. Importantly, a replay ratio of 4 eclipses the results at a ratio of 2, highlighting the approach's scalability.

Collectively, these experiments reiterate our method's generality and its alignment with modern algorithms.

[2] Bigger, Better, Faster: Human-level Atari with human-level efficiency. Schwarzer et al., ICML 2023.

[3] On the Importance of Feature Decorrelation for Unsupervised Representation Learning for RL., ICML 2023.


> **Extension to DMC Generalization Benchmark**

Diverging from the conventional DMC-GB-500k benchmark [4], we opted for a consistently noisy environment both for training and testing. This choice was taken to evaluate generalization techniques under persistent input non-stationarity. Here, we used 5 different environments, {walker-walk, cart pole-swing-up, finger-spin, walker-stand, ball-in-cup-catch}.

Table A.3 encapsulates our findings:

SAM, when paired with reset, exhibits a performance closely mirroring that of Adam. However, without the influence of a reset, SAM surpasses Adam.

Contrary to our initial presumptions, we found that the DMC environment manifested a pronounced degree of input non-stationarity in contrast to Atari, even when excluding DMC-GB's intrinsic noise. This observation leads to the insight that "DMC inherently calls for methods addressing input non-stationarity."

Such a realization prompted a deeper introspection into the role of reset strategy. As detailed in the original reset paper [5], DMC employs a notably aggressive reset strategy — engaging up to 90% reset for the DrQ algorithm and a comprehensive reset for SAC. Meanwhile, Atari adopts a milder reset approach, reinitializing around half of its network. DMC's rigorous reset approach serves as an effective antidote to its inherent input non-stationarity. This might explain why the benefits of SAM appear subdued when combined with such an assertive reset. Intriguingly, our evaluations on synthetic and Atari platforms revealed that Reset (B, H) strategies faltered in performance. This highlights the necessity for a more tailored approach, especially as network sizes expand.

To summarize, we believe effectively managing input non-stationarity in RL is a key to improving performance for both DMC and DMC-GB.

[4] Generalization in Reinforcement Learning by Soft Data Augmentation., Hansen et al., ICRA 2021
[5] The Primacy Bias in Deep Reinforcement Learning., Nikishin et al., ICML 2022

> **Conclusion**

In conclusion, we deeply appreciate the constructive feedback from the reviewers. Their insights have refined our research, and we believe our revisions now thoroughly address the raised concerns. We eagerly await further feedback.

Warm regards,
Authors of Submission 12455.

---

### Decision · Program_Chairs · 2023-09-21

**Decision:**

Accept (poster)

**Comment:**

Clear accept.

This paper investigates improving sample and computational efficiency in RL via a new method that combines periodic resets and Sharpness-Aware Minimization (two know ideas, but SAM is new to RL) and showing both are needed. The reviewers found the method simple and appealing, and the general thesis interesting and well motivated.

Experiments show the new method helps when applied in supervised learning and when combined with two recent, popular RL methods in Atari 100k benchmark and Deepmind Control Suite. Reviewer GE2W pointed out major issues with the experiments, and the authors heroically made the requested changes (including increasing # of seeds from 5 to 30) and GE2W was happy; although this changed how the results should be discussed (more on that below). Reviewer mzhA had similar requests for additional results and clarifications and the authors provided them.

All reviewers voted to accept, and two domain experts voted clear accept. The reviewers noted the paper was well written and nicely polished with measured claims and well executed experiments.

Please ensure the following changes are complete for camera ready:
- do not bold results in tables where the errorbars (+/-) overlap. This is misleading
- Reviewers dLhS and GE2W gave advice on how to adjust the text based on the new results (e.g., some results are now a statistical tie which is ok if accurately and honestly discussed). Please implement these writing adjustments to the abstract, intro and results discussion.